# Improving Generalization and Data Efficiency with Diffusion in Offline Multi-agent RL

**Zhuoran Li**                                                     *lizr20@mails.tsinghua.edu.cn*
*Institute for Interdisciplinary Information Sciences*
*Tsinghua University*
**Ling Pan**                                                               *lingpan@ust.hk*
*Department of Electronic and Computer Engineering*
*Hong Kong University of Science and Technology*
**Jiatai Huang**                                                   *hjt18@mails.tsinghua.edu.cn*
*Independent Researcher*
**Longbo Huang**\*                                             *longbohuang@tsinghua.edu.cn*
*Institute for Interdisciplinary Information Sciences*
*Tsinghua University*

**Reviewed on OpenReview:** *https://openreview.net/forum?id=GKuCKSJKvl*

## Abstract

We present a novel Diffusion Offline Multi-agent Model (DOM2) for offline Multi-Agent Reinforcement Learning (MARL). Different from existing algorithms that rely mainly on conservatism in policy design, DOM2 enhances policy expressiveness and diversity based on diffusion model. Specifically, we incorporate a diffusion model into the policy network and propose a trajectory-based data-reweighting scheme in training. These key ingredients significantly improve algorithm robustness against environment changes and achieve significant improvements in performance, generalization and data-efficiency. Our extensive experimental results demonstrate that DOM2 outperforms existing state-of-the-art methods in all multi-agent particle and multi-agent MuJoCo environments, and generalizes significantly better to shifted environments (in 28 out of 30 settings evaluated) thanks to its high expressiveness and diversity. Moreover, DOM2 is ultra data efficient and requires no more than 5% data for achieving the same performance compared to existing algorithms (a 20× improvement in data efficiency).

## 1 Introduction

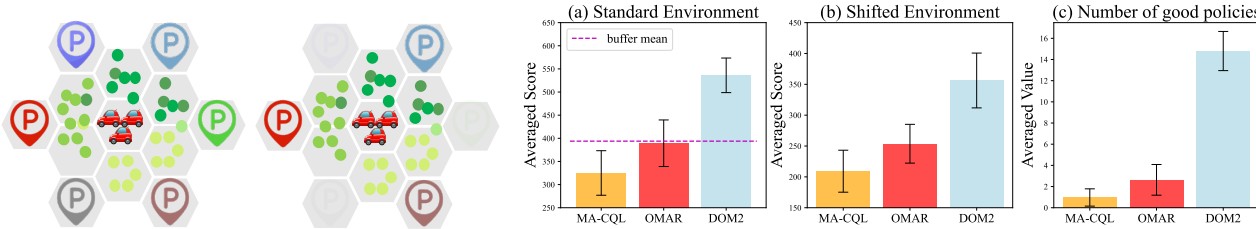

Figure 1: Left: Standard environment (left) and shifted environment dismissing 3 landmarks randomly (right). Right: (a) Results in standard environments. (b) Results in shifted environments. (c) Number of good policies in standard environments. For experimental details, see Appendix A.2.4.

---

\*Corresponding Author: Longbo Huang. Email: longbohuang@tsinghua.edu.cn

Offline reinforcement learning (RL), commonly referred to as batch RL, aims to learn efficient policies exclusively from previously gathered data without interacting with the environment (Lange et al., 2012; Levine et al., 2020). Since the agent has to sample the data from a fixed dataset, naive offline RL approaches fail to learn policies for out-of-distribution actions or states (Wu et al., 2019; Kumar et al., 2019), and the obtained Q-value estimation for these actions will be inaccurate with unpredictable consequences. Recent progress in tackling the problem focuses on *conservatism* by introducing regularization terms to constrain the policy and Q-value training, e.g, penalizing the Q-values of the unseen actions (Fujimoto et al., 2019; Kumar et al., 2020a; Fujimoto & Gu, 2021; Kostrikov et al., 2021a; Lee et al., 2022). These conservatism-based offline RL algorithms have achieved significant progress in difficult offline multi-agent reinforcement learning settings (MARL) (Jiang & Lu, 2021; Yang et al., 2021; Pan et al., 2022).

Despite the potential benefits, existing methods have limitations in several aspects. Firstly, the design of the policy network and the corresponding regularizer limits the expressiveness and diversity. Consequently, the resulting policy may be suboptimal and fail to represent complex strategies, e.g., policies with multi-modal distribution over actions (Kumar et al., 2019; Wang et al., 2022). Secondly, in multi-agent scenarios, the conservatism-based method is prone to getting trapped in poor local optima. This occurs when each agent is incentivized to maximize its own reward without efficient cooperation with other agents in existing algorithms (Yang et al., 2021; Pan et al., 2022). To demonstrate this phenomenon, we conduct experiment on a simple MARL scenario consisting of 3 agents and 6 landmarks (Figure 1a), to highlight the importance of policy expressiveness and diversity in MARL. In this scenario, the agents are asked to cover 3 landmarks and are rewarded based on their proximity to the nearest landmark while being penalized for collisions. We first train the agents with 6 target

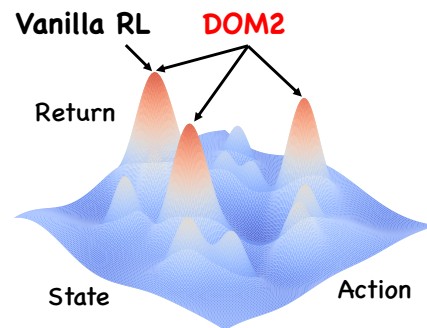

Figure 2: Schematic to illustrate that DOM2 obtains better diversity. When multiple actions yield near-optimal rewards, DOM2 can uncover more such solutions than standard offline MARL algorithms.

landmarks and then randomly dismiss 3 of them in evaluation. Our experiments demonstrate that existing methods (MA-CQL and OMAR (Pan et al., 2022)), which constrain policies through regularization, limit the expressiveness of each agent and hinder the ability of the agents to cooperate with diversity. As a result, only limited solutions are found. Therefore, in order to design robust algorithms with good generalization capabilities, it is crucial to develop novel methods for better performance and more efficient cooperation among agents.

To boost the policy expressiveness and diversity, we propose a novel algorithm based on *diffusion* for the offline multi-agent setting, called Diffusion Offline Multi-Agent Model (DOM2). Diffusion model has shown significant success in generating data with high quality and diversity (Ho et al., 2020; Song et al., 2020b; Wang et al., 2022; Croitoru et al., 2023). Our goal is to leverage this advantage to promote expressiveness and diversity of the policy. In Figure 2, we emphasize a key message: the diffusion-based policy has the potential to find more well-performing solutions compared to general offline reinforcement learning methods. This advantage arises from the diffusion model's ability to more effectively capture multi-modal distributions, enabling a richer representation of the underlying policy landscape.

However, a crucial challenge arises that the training objectives of diffusion model and offline reinforcement learning are inconsistent. In order to train an appropriate policy that performs well, we propose a trajectory-based data-reweighting method to facilitate policy training by efficient data sampling. For efficient sampling, the policy for each agent is built using the accelerated DPM-solver to sample actions (Lu et al., 2022). These techniques enable the policy to generate solutions with high quality and diversity, and overcome the aforementioned limitations. Figure 1b shows that in the 3-agent example, DOM2 can find a more diverse set of solutions with high performance and generalization, compared to conservatism-based methods such as MA-CQL and OMAR (Pan et al., 2022). Our contributions are summarized as follows.

- We propose a novel Diffusion Offline Multi-Agent Model (DOM2) algorithm to address the limitations of conservatism-based methods. DOM2 is a decentralized training and execution framework consisting of three critical components: a diffusion-based policy with an accelerated solver (sampling within 10 steps), an appropriate policy regularizer, and a trajectory-based data reweighting method for enhancing learning.

- We conduct extensive numerical experiments on Multi-agent Particles Environments (MPE) and Multi-agent MuJoCo (MAMuJoCo) HalfCheetah environments. Our results show that DOM2 achieves significantly better performance improvement over state-of-the-art methods in all tasks.

- We show that DOM2 possesses much better generalization abilities and outperforms existing methods in shifted environments, i.e., DOM2 achieves state-of-the-art performance in 17 out of 18 MPE shifted settings and 11 out of 12 MAMuJoCo shifted settings. Moreover, DOM2 is ultra-data-efficient and achieves SOTA performance with $20\times$ times less data.

## 2 Related Work

We discuss a set of related work about the offline RL, MARL and diffusion models.

**Offline RL and MARL:** Distribution shift is a key obstacle in offline RL and multiple methods have been proposed to tackle the problem based on conservatism to constrain the policy or Q-value by regularizers (Wu et al., 2019; Kumar et al., 2019; Fujimoto et al., 2019; Kumar et al., 2020a). Policy regularization ensures the policy to be close to the behavior policy via a policy regularizer, e.g., BRAC (Wu et al., 2019), BEAR (Kumar et al., 2019), BCQ (Fujimoto et al., 2019), TD3+BC (Fujimoto & Gu, 2021), implicit update (Peng et al., 2019; Siegel et al., 2020; Nair et al., 2020) and importance sampling (Kostrikov et al., 2021a; Swaminathan & Joachims, 2015; Liu et al., 2019; Nachum et al., 2019)). Critic regularization instead constrains the Q-values for stability, e.g., CQL (Kumar et al., 2020a), IQL (Implicit Q-Learning) (Kostrikov et al., 2021b), and TD3-CVAE (Rezaeifar et al., 2022). On the other hand, Multi-Agent Reinforcement Learning (MARL) has made significant process, such as MADDPG (Lowe et al., 2017), MAPPO (Yu et al., 2021), VDN (Sunehag et al., 2017) and QMIX (Rashid et al., 2018) under the centralized training with decentralized execution (CTDE) paradigm (Oliehoek et al., 2008; Matignon et al., 2012), and IQL (Independent Q-Learning) (Tampuu et al., 2017), MATD3 (Ackermann et al., 2019) and IPPO (de Witt et al., 2020) are designed as fully decentralized training and execution scheme. The offline MARL problem has also attracted attention using conservatism-based methods, e.g., MA-BCQ (Jiang & Lu, 2021), MA-ICQ (Yang et al., 2021), MA-CQL, OMAR (Pan et al., 2022) and CFCQL (Shao et al., 2023).

**Diffusion Models:** Diffusion model (Ho et al., 2020; Song et al., 2020b; Sohl-Dickstein et al., 2015; Song & Ermon, 2019; Song et al., 2020a), a specific type of generative model, has shown significant success in various applications, especially in generating images from text descriptions (Nichol et al., 2021; Ramesh et al., 2022; Saharia et al., 2022)). Recent works have focused on the foundation of diffusion models, e.g., the statistical theory (Chen et al., 2023), and the accelerating method for sampling (Lu et al., 2022; Bao et al., 2022). Generative model has been applied to policy modeling, including conditional VAE (Kingma & Welling, 2013), diffusers (Janner et al., 2022; Ajay et al., 2022), Diffusion-QL (Wang et al., 2022), SfBC (Chen et al., 2022; Lu et al., 2023a), IDQL (Hansen-Estruch et al., 2023) and DAC (Fang et al., 2024) in the single-agent setting and MA-DIFF (Zhu et al., 2023) and DoF (Li et al., 2025) in multi-agent setting.

We emphasize that most existing works focus on conservatism for algorithm design, i.e., CQL (Kumar et al., 2020a) constrains the Q-value into a safe range and OMAR (Pan et al., 2022) constrains the policy network to sample actions with rectification technique to approach the seen actions in the dataset. Our algorithm goes beyond this and focuses on introducing diffusion into offline MARL with the accelerated solver under fully decentralized training and execution structure.

## 3 Background

In this section, we introduce the offline multi-agent reinforcement learning problem and provide preliminaries for the diffusion probabilistic model as the background for our proposed algorithm.

**Offline Multi-Agent Reinforcement Learning.** A fully cooperative multi-agent task can be modeled as a decentralized partially observable Markov decision process (Dec-POMDP (Oliehoek & Amato, 2016)) with $n$ agents consisting of a tuple $G = \langle \mathcal{I}, \mathcal{S}, \mathcal{O}, \mathcal{A}, \Pi, \mathcal{P}, \mathcal{R}, n, \gamma \rangle$. Here $\mathcal{I}$ is the set of agents, $\mathcal{S}$ is the global state space, $\mathcal{O} = (\mathcal{O}_1, ..., \mathcal{O}_n)$ is the set of observations with $\mathcal{O}_n$ being the set of observation for agent $n$. $\mathcal{A} = (\mathcal{A}_1, ..., \mathcal{A}_n)$ is the set of actions for the agents ($\mathcal{A}_n$ is the set of actions for agent $n$), $\Pi = (\Pi_1, ..., \Pi_n)$ is the set of policies, and $\mathcal{P}$ is the function class of the transition probability $\mathcal{S} \times \mathcal{A} \times \mathcal{S}' \to [0, 1]$. At each time step $t$, each agent chooses an action $a_j^t \in \mathcal{A}_j$ based on the policy $\pi_j \in \Pi_j$ and historical observation $o_j^{t-1} \in \mathcal{O}_j$. The next state is determined by the transition probability $P \in \mathcal{P}$. Each agent then receives a reward $r_j^t \in \mathcal{R} : \mathcal{S} \times \mathcal{A} \to \mathbb{R}$ and a private observation $o_j^t \in \mathcal{O}_i$. The goal of the agents is to find the optimal policies $\boldsymbol{\pi} = (\pi_1, ..., \pi_n)$ such that each agent can maximize the discounted return: $\mathbb{E}[\sum_{t=0}^{\infty} \gamma^t r_j^t]$ (the joint discounted return is $\mathbb{E}[\sum_{j=1}^{n} \sum_{t=0}^{\infty} \gamma^t r_j^t]$), where $\gamma$ is the discount factor. Offline reinforcement learning requires that the data to train the agents is sampled from a given dataset $\mathcal{D}$ generated from some potentially unknown behavior policy $\boldsymbol{\pi}_{\boldsymbol{\beta}}$ (which can be arbitrary). This means that the procedure for training agents is separated from the interaction with environments.

**Conservative Q-Learning.** For training the critic in offline RL, the conservative Q-Learning (CQL) method (Kumar et al., 2020a) is to train the Q-value function $Q_{\boldsymbol{\phi}}(\boldsymbol{o}, \boldsymbol{a})$ parameterized by $\boldsymbol{\phi}$, by minimizing the temporal difference (TD) loss plus the conservative regularizer. Specifically, the objective to optimize the Q-value for each agent $j$ is given by:

$$\mathcal{L}(\boldsymbol{\phi}_j) = \mathbb{E}_{(\boldsymbol{o}_j, \boldsymbol{a}_j) \sim \mathcal{D}_j}[(y_j - Q_{\boldsymbol{\phi}_j}(\boldsymbol{o}_j, \boldsymbol{a}_j))^2] + \zeta \mathbb{E}_{(\boldsymbol{o}_j, \boldsymbol{a}_j) \sim \mathcal{D}_j}[\log \sum_{\tilde{\boldsymbol{a}}_j} \exp(Q_{\boldsymbol{\phi}_j}(\boldsymbol{o}_j, \tilde{\boldsymbol{a}}_j)) - Q_{\boldsymbol{\phi}_j}(\boldsymbol{o}_j, \boldsymbol{a}_j)]. \tag{1}$$

The first term is the TD error to minimize the Bellman operator with the double Q-learning trick (Fujimoto et al., 2019; Hasselt, 2010; Lillicrap et al., 2015), where $y_j = r_j + \gamma \min_{k=1,2} \overline{Q}_{\boldsymbol{\phi}_j}^k(\boldsymbol{o}_j', \pi_j(\boldsymbol{o}_j'))$, $\overline{Q}_{\overline{\boldsymbol{\phi}}_j}, \overline{\pi}_j$ denotes the target network and $\boldsymbol{o}_j'$ is the next observation for agent $j$ after taking action $\boldsymbol{a}_j$. The second term is a conservative regularizer, where $\tilde{\boldsymbol{a}}_j$ is a random action uniformly sampled in the action space and $\zeta$ is a hyperparameter to balance two terms. The regularizer is to address the extrapolation error by encouraging large Q-values and penalizing low Q-values for state-action pairs in the dataset.

**Diffusion Probabilistic Model.** We present a high-level introduction to the Diffusion Probabilistic Model (DPM) (Sohl-Dickstein et al., 2015; Song et al., 2020b; Ho et al., 2020) (detailed introduction is in Appendix A.1). DPM is a deep generative model that learns the unknown data distribution $\boldsymbol{x}_0 \sim q_0(\boldsymbol{x}_0)$ from the dataset. DPM has a predefined forward noising process characterized by a stochastic differential equation (SDE) $d\boldsymbol{x}_t = f(t)\boldsymbol{x}_t dt + g(t) d\boldsymbol{w}_t$ (Equation (5) in (Song et al., 2020b)) and a trainable reverse denoising pro-

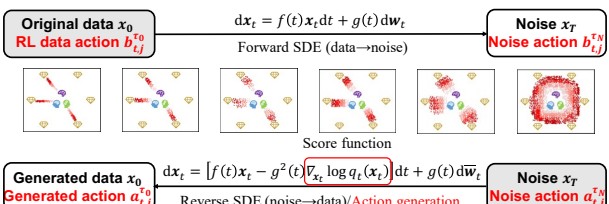

Figure 3: Diffusion probabilistic model as a stochastic differential equation (SDE) (Song et al., 2020b) and relationship with Offline MARL.

cess characterized by the SDE $d\boldsymbol{x}_t = [f(t)\boldsymbol{x}_t - g^2(t)\nabla_{\boldsymbol{x}_t} \log q_t(\boldsymbol{x}_t)]dt + g(t)d\overline{\boldsymbol{w}}_t$ (Equation (6) in (Song et al., 2020b)) shown in Figure 3. Here $\boldsymbol{w}_t, \overline{\boldsymbol{w}}_t$ are standard Brownian motions, $f(t), g(t)$ are pre-defined functions such that $q_{0t}(\boldsymbol{x}_t|\boldsymbol{x}_0) = \mathcal{N}(\boldsymbol{x}_t; \alpha_t \boldsymbol{x}_0, \sigma_t^2 \boldsymbol{I})$ for some constant $\alpha_t, \sigma_t > 0$ and $q_T(\boldsymbol{x}_T) \approx \mathcal{N}(\boldsymbol{x}_T; \boldsymbol{0}, \tilde{\sigma}^2 \boldsymbol{I})$ is almost a Gaussian distribution for constant $\tilde{\sigma} > 0$. However, there exists an unknown term $\nabla_{\boldsymbol{x}_t} \log q_t(\boldsymbol{x}_t)$, which is called the *score function* (Song et al., 2020a). In order to generate data close to the distribution $q_0(\boldsymbol{x}_0)$ by the reverse SDE, DPM defines a score-based model $\boldsymbol{\epsilon}_{\boldsymbol{\theta}}(\boldsymbol{x}_t, t)$ to learn the score function and optimize parameter $\boldsymbol{\theta}$ such that $\boldsymbol{\theta}^* = \arg \min_{\boldsymbol{\theta}} \mathbb{E}_{\boldsymbol{x}_0 \sim q_0(\boldsymbol{x}_0), \boldsymbol{\epsilon} \sim \mathcal{N}(\boldsymbol{0}, \boldsymbol{I}), t \sim \mathcal{U}(0,T)}[\|\boldsymbol{\epsilon} - \boldsymbol{\epsilon}_{\boldsymbol{\theta}}(\alpha_t \boldsymbol{x}_0 + \sigma_t \boldsymbol{\epsilon}, t)\|_2^2]$ ($\mathcal{U}(0, T)$ is the uniform distribution in $[0, T]$, same later). With the learned score function, we can sample data by discretizing the reverse SDE. To enable faster sampling, DPM-solver (Lu et al., 2022) provides an efficiently faster sampling method and the first-order iterative equation (Equation (3.7) in (Lu et al., 2022)) to denoise is given by $\boldsymbol{x}_{t_i} = \frac{\alpha_{t_i}}{\alpha_{t_{i-1}}} \boldsymbol{x}_{t_{i-1}} - \sigma_{t_i}(\frac{\alpha_{t_i} \sigma_{t_{i-1}}}{\alpha_{t_{i-1}} \sigma_{t_i}} - 1)\boldsymbol{\epsilon}_{\boldsymbol{\theta}}(\boldsymbol{x}_{t_{i-1}}, t_{i-1})$.

In Figure 3, we highlight a crucial message that we can efficiently incorporate the procedure of data generation into offline MARL as the action generator. Intuitively, we can utilize the fixed dataset to learn an action generator by noising the sampled actions in the dataset, and then denoising it inversely. The procedure

assembles data generation in the diffusion model. However, it is important to note that there is a critical difference between the objectives of diffusion and RL. Specifically, in diffusion model, the goal is to generate data with a distribution close to the distribution of the training dataset, whereas in offline MARL, one hopes to find actions (policies) that maximize the joint discounted return. This difference influences the design of the action generator. Properly handling it is the key in our design, which will be detailed below in Section 4.

## 4 Proposed Method

In this section, we present the DOM2 algorithm shown in Figure 4. In the following, we first discuss how we generate the actions with diffusion in Section 4.1. Next, we show how to design appropriate objective functions in policy learning in Section 4.2. We then present the data reweighting method in Section 4.3. Finally, we present the whole procedure of DOM2 in Section 4.4.

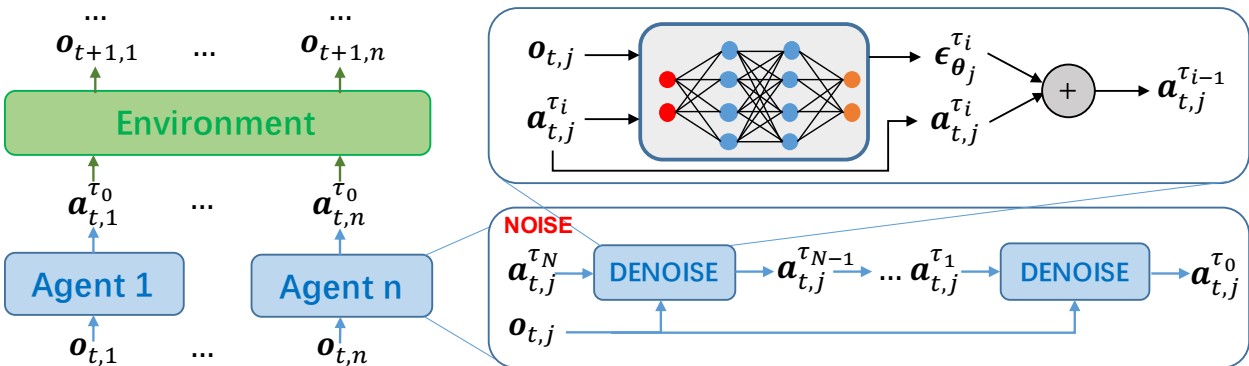

Figure 4: Diagram of the DOM2 algorithm. Each agent generates actions with diffusion.

### 4.1 Diffusion in Offline MARL

We first present the diffusion component in DOM2, which generates actions by denoising a Gaussian noise iteratively (shown on the right side of Figure 4). Denote the timestep indices in an episode by $\{t\}_{t=1}^{T}$, the diffusion step indices by $\tau \in [\tau_0, \tau_N]$, and the agent by $\{j\}_{j=1}^{n}$. Below, to facilitate understanding, we introduce the diffusion idea in continuous time, based on (Song et al., 2020b; Lu et al., 2022). We then present our algorithm design by specifying the discrete DPM-solver-based steps (Lu et al., 2022) and discretizing diffusion timesteps, i.e., from $[\tau_0, \tau_N]$ to $\{\tau_i\}_{i=0}^{N}$.

(**Noising**) Noising the action in diffusion is modeled as a forward process from $\tau_0$ to $\tau_N$. Specifically, we start with the collected action data at $\tau_0$, denoted by $\boldsymbol{b}_{t,j}^{\tau_0} \sim \boldsymbol{\pi}_{\beta_j}(\cdot|\boldsymbol{o}_{t,j})$, which is collected from the behavior policy $\boldsymbol{\pi}_{\beta_j}(\cdot|\boldsymbol{o}_{t,j})$. We then perform a set of noising operations on intermediate data $\{\boldsymbol{b}_{t,j}^{\tau}\}_{\tau \in [\tau_0, \tau_N]}$, and eventually generate $\boldsymbol{b}_{t,j}^{\tau_N}$, which (ideally) is close to Gaussian noise at $\tau_N$. This forward process satisfies that for $\forall \tau \in [\tau_0, \tau_N]$, the transition probability $q_{\tau_0\tau}(\boldsymbol{b}_{t,j}^{\tau}|\boldsymbol{b}_{t,j}^{\tau_0}) = \mathcal{N}(\boldsymbol{b}_{t,j}^{\tau}; \alpha_\tau \boldsymbol{b}_{t,j}^{\tau_0}, \sigma_\tau^2 \boldsymbol{I})$ (Lu et al., 2022). The selection of the noise schedules $\alpha_\tau, \sigma_\tau$ enables that $q_{\tau_N}(\boldsymbol{b}_{t,j}^{\tau_N}|\boldsymbol{o}_{t,j}) \approx \mathcal{N}(\boldsymbol{b}_{t,j}^{\tau_N}; \boldsymbol{0}, \tilde{\sigma}^2 \boldsymbol{I})$ for some $\tilde{\sigma} > 0$, which is almost a Gaussian noise. According to (Song et al., 2020b; Kingma et al., 2021), there exists a corresponding reverse process of SDE from $\tau_N$ to $\tau_0$ (based on Equation (2.4) in (Lu et al., 2022)) considering $\boldsymbol{o}_{t,j}$ as conditions:

$$\mathrm{d}\boldsymbol{a}_{t,j}^{\tau} = [f(\tau)\boldsymbol{a}_{t,j}^{\tau} - g^2(\tau)\underbrace{\nabla_{\boldsymbol{b}_{t,j}^{\tau}} q_\tau(\boldsymbol{b}_{t,j}^{\tau}|\boldsymbol{o}_{t,j})}_{\text{Neural Network } \boldsymbol{\epsilon}_{\boldsymbol{\theta}_j}}]\mathrm{d}\tau + g(\tau)\mathrm{d}\overline{\boldsymbol{w}}_\tau,$$

(2)

where $f(\tau) = \frac{\mathrm{d}\log\alpha_\tau}{\mathrm{d}\tau}, g^2(\tau) = \frac{\mathrm{d}\sigma_\tau^2}{\mathrm{d}\tau} - 2\frac{\mathrm{d}\log\alpha_\tau}{\mathrm{d}\tau}\sigma_\tau^2$ and $\overline{\boldsymbol{w}}_t$ is a Brownian motion, $\boldsymbol{a}_{t,j}^{\tau_N} \sim q_{\tau_N}(\boldsymbol{b}_{t,j}^{\tau_N}|\boldsymbol{o}_{t,j})$, and $\boldsymbol{a}_{t,j}^{\tau_0}$ is the generated action for agent $j$ at time $t$. To fully determine the reverse process of SDE described by equation 2, we need the access to the scaled conditional *score function* $-\sigma_\tau \nabla_{\boldsymbol{b}_{t,j}^{\tau}} q_\tau(\boldsymbol{b}_{t,j}^{\tau}|\boldsymbol{o}_{t,j})$ at each $\tau$.

We use a neural network $\boldsymbol{\epsilon}_{\boldsymbol{\theta}_j}(\boldsymbol{b}_{t,j}^{\tau}, \boldsymbol{o}_{t,j}, \tau)$ to represent it and the architecture is the multiple-layered residual network, which is shown in Figure 9 that resembles U-Net (Ho et al., 2020; Chen et al., 2022). The objective of optimizing the parameter $\boldsymbol{\theta}_j$ is (based on (Lu et al., 2022)):

$$\mathcal{L}_{bc}(\boldsymbol{\theta}_j) = \mathbb{E}_{(\boldsymbol{o}_{t,j}, \boldsymbol{a}_{t,j}^{\tau_0}) \sim \mathcal{D}_j, \boldsymbol{\epsilon} \sim \mathcal{N}(\mathbf{0}, \boldsymbol{I}), \tau \in \mathcal{U}(\{\tau_i\}_{i=0}^N)}[\|\boldsymbol{\epsilon} - \boldsymbol{\epsilon}_{\boldsymbol{\theta}_j}(\alpha_\tau \boldsymbol{a}_{t,j}^{\tau_0} + \sigma_\tau \boldsymbol{\epsilon}, \boldsymbol{o}_{t,j}, \tau)\|_2^2]. \tag{3}$$

(**Denoising**) After training the neural network $\boldsymbol{\epsilon}_{\boldsymbol{\theta}_j}$, we can then generate the actions by solving the diffusion SDE in equation 2 (plugging in $-\boldsymbol{\epsilon}_{\boldsymbol{\theta}_j}(\boldsymbol{a}_{t,j}^{\tau}, \boldsymbol{o}_{t,j}, \tau)/\sigma_\tau$ to replace the true score function $\nabla_{\boldsymbol{b}_{t,j}^{\tau}} \log q_\tau(\boldsymbol{b}_{t,j}^{\tau}|\boldsymbol{o}_{t,j}))$. Here we evolve the reverse process of SDE from $\boldsymbol{a}_{t,j}^{\tau_N} \sim \mathcal{N}(\boldsymbol{a}_{t,j}^{\tau_N}; \mathbf{0}, \boldsymbol{I})$, a Gaussian noise, and we take $\boldsymbol{a}_{t,j}^{\tau_0}$ as the final action. To facilitate faster sampling, we discretize the reverse process of SDE in $[\tau_0, \tau_N]$ into $N+1$ diffusion timesteps $\{\tau_i\}_{i=0}^N$ (the partition details are shown in Appendix A.1) and adopt the first-order DPM-solver-based method (Equation (3.7) in (Lu et al., 2022)) to iteratively denoise from $\boldsymbol{a}_{t,j}^{\tau_N} \sim \mathcal{N}(\boldsymbol{a}_{t,j}^{\tau_N}; \mathbf{0}, \boldsymbol{I})$ to $\boldsymbol{a}_{t,j}^{\tau_0}$ for $i = N, ..., 1$ written as:

$$\boldsymbol{a}_{t,j}^{\tau_{i-1}} = \frac{\alpha_{\tau_{i-1}}}{\alpha_{\tau_i}} \boldsymbol{a}_{t,j}^{\tau_i} - \sigma_{\tau_i}\left(\frac{\alpha_{\tau_i}\sigma_{\tau_{i-1}}}{\alpha_{\tau_{i-1}}\sigma_{\tau_i}} - 1\right)\boldsymbol{\epsilon}_{\boldsymbol{\theta}_j}(\boldsymbol{a}_{t,j}^{\tau_i}, \boldsymbol{o}_{t,j}, \tau_i), \tag{4}$$

for $i = N, ... 1$, and the iterative denoising steps correspond to the right panel of Figure 4.

## 4.2 Policy Improvement

Notice that optimizing $\boldsymbol{\theta}_j$ solely using equation 3 is insufficient in offline MARL, because it will not be able to generate actions beyond the behavior policy. To achieve policy improvement, inspired by (Wang et al., 2022), we use the following loss function involving both the diffusion term and the Q-value:

$$\mathcal{L}(\boldsymbol{\theta}_j) = \mathcal{L}_{bc}(\boldsymbol{\theta}_j) + \mathcal{L}_q(\boldsymbol{\theta}_j) = \mathcal{L}_{bc}(\boldsymbol{\theta}_j) - \tilde{\eta}\mathbb{E}_{(\boldsymbol{o}_j, \boldsymbol{a}_j)\sim\mathcal{D}_j, \boldsymbol{a}_j^{\tau_0}\sim\pi_{\boldsymbol{\theta}_j}}[Q_{\boldsymbol{\phi}_j}(\boldsymbol{o}_j, \boldsymbol{a}_j^{\tau_0})]. \tag{5}$$

The second term $\mathcal{L}_q(\boldsymbol{\theta}_j)$ is called Q-loss (Wang et al., 2022) for policy improvement , where $\boldsymbol{a}_j^{\tau_0}$ is generated by equation 4, $\boldsymbol{\phi}_j$ is the network parameter of Q-value function for agent $j$, $\tilde{\eta} = \frac{\eta}{\mathbb{E}_{(\boldsymbol{s}_j, \boldsymbol{a}_j)\sim\mathcal{D}}[Q_{\boldsymbol{\phi}_j}(\boldsymbol{o}_j, \boldsymbol{a}_j)]}$ and $\eta$ is a hyperparameter. This Q-value is normalized to control the scale of Q-value functions (Fujimoto & Gu, 2021) and $\eta$ is used to balance the weights. The combination of two terms ensures that the policy can preferentially sample actions with high values. The reason is that the policy trained by optimizing equation 5 can generate actions with different distributions compared to the behavior policy, and the policy prefers to sample actions with higher Q-values (corresponding to better performance). To train efficient Q-values for policy improvement, we optimize equation 1 as the objective (Kumar et al., 2020a).

## 4.3 Data Reweighting

In DOM2, in addition to the novel policy design with its training objectives, we also introduce a data-reweighting method to scale up the size of the dataset. Specifically, we replicate trajectories $\mathcal{T}_i \in \mathcal{D}$ with high return values (i.e., with the return value, denoted by $Return(\mathcal{T}_i)$, higher than threshold values) in the dataset. Specifically, we define a set of threshold values $\mathcal{R} = \{r_{\text{th},1}, ..., r_{\text{th},K}\}$. Then, we compare the reward of each trajectory with every threshold value and replicate the trajectory once whenever its return is higher than the compared threshold (Line 3 in Algorithm 1, which will be introduced below), such that trajectories with higher returns can replicate more times. Doing so allows us to create more data efficiently and improve the performance of the policy by increasing the probability of sampling trajectories with better performance in the dataset. We emphasize that our method is different from the data augmented works, where the objective is to use a diffusion model as a data generator for downstream tasks, e.g., (Trabucco et al., 2023; Lu et al., 2023b). Our method is designed to enhance the offline dataset for facilitating diffusion-based policy and Q-value training in offline MARL.

## 4.4 The DOM2 Algorithm and Discussions

The resulting DOM2 algorithm is presented in Algorithm 1. Line 1 is the initialization step. Line 3 is the data-reweighting step. Line 7 is the sampling procedure for the preparation of the mini-batch data from

the replicated dataset to train the agents. Lines 8 and 9 are the update of actor and critic parameters, i.e., the policy and the Q-value. Line 10 is the soft update procedure for the target networks. Our algorithm provides a systematic way to integrate diffusion into RL algorithm with appropriate regularizers and how to train the diffusion policy in a decentralized multi-agent setting.

The proposed DOM2 algorithm establishes an innovative framework for generating actions in offline multi-agent reinforcement learning (MARL) through a diffusion model-based policy network. This framework integrates diffusion model-based losses with reinforcement learning-based Q-value functions to optimize the policy network. Sampling efficiency is improved by employing accelerated solvers, while data efficiency is enhanced via data reweighting strategies. This offline, decentralized learning framework addresses the limitations of previous conservative methods, which often failed to identify an adequate set of well-performing policies. By leveraging the diffusion model's enhanced capability to approximate complex data distributions, the DOM2 framework uncovers a broader range of desirable solutions for offline MARL problems. This innovation not only enhances performance across base environments but also improves the algorithm's generalization ability. The DOM2 algorithm's capability to identify a diverse set of high-performing policies further contributes to its robustness against environmental variability, thereby enhancing generalization, defined as the performance of the algorithm in previously unseen environments. Additionally, the diffusion model-based policy network architecture and data reweighting methodology significantly bolster data efficiency, which is reflected as the algorithm's performance under limited data conditions. These advantages will be further analyzed and demonstrated in detail in the experimental results.

---

**Algorithm 1** Diffusion Offline Multi-agent Model (`DOM2`) Algorithm

1: **Input:** Initialize Q-networks $Q^1_{\phi_j}, Q^2_{\phi_j}$, policy network $\pi_j$ with random parameters $\phi^1_j, \phi^2_j, \boldsymbol{\theta}_j$, target networks with $\overline{\phi}^1_j \leftarrow \phi^1_j, \overline{\phi}^2_j \leftarrow \phi^2_j, \overline{\boldsymbol{\theta}}_j \leftarrow \boldsymbol{\theta}_j$ for each agent $j = 1, \dots, N$, dataset $\mathcal{D}$ with trajectories $\{\mathcal{T}_i\}_{i=1}^L$ and replicated dataset $\mathcal{D}' \leftarrow \mathcal{D}$. // Initialization
2: **for** every $r_{\text{th}} \in \mathcal{R}$ **do**
3:     $\mathcal{D}' \leftarrow \mathcal{D}' + \{\mathcal{T}_i \in \mathcal{D} | Return(\mathcal{T}_i) \geq r_{\text{th}}\}$. // Data Reweighting
4: **end for**
5: **for** training step $t = 1$ **to** $T$ **do**
6:     **for** agent $j = 1$ **to** $n$ **do**
7:         Sample a random minibatch of $\mathcal{S}$ samples $(\boldsymbol{o}_j, \boldsymbol{a}_j, \boldsymbol{r}_j, \boldsymbol{o}'_j)$ from dataset $\mathcal{D}'$. // Sampling
8:         Update critics $\phi^1_j, \phi^2_j$ to minimize equation 1. // Update Critic
9:         Update the actor $\boldsymbol{\theta}_j$ to minimize equation 5. // Update Actor with Diffusion
10:         Update target networks: $\overline{\phi}^k_j \leftarrow \rho\phi^k_j + (1-\rho)\overline{\phi}^k_j, (k=1,2), \overline{\boldsymbol{\theta}}_j \leftarrow \rho\boldsymbol{\theta}_j + (1-\rho)\overline{\boldsymbol{\theta}}_j$.
11:     **end for**
12: **end for**

---

Some comparisons with the recent diffusion-based methods for action generation are in place. First of all, we use the diffusion-based policy in the multi-agent setting. Then, different from Diffuser (Janner et al., 2022), our method generates actions independently among different timesteps, while Diffuser generates a sequence of actions as a trajectory in the episode using a combination of diffusion model and the transformer architecture, so the actions are dependent among different timesteps. Compared to the DDPM-based diffusion policy (Wang et al., 2022), we use the first-order DPM-Solver (Lu et al., 2022) and the multi-layer residual network as the noise network (Chen et al., 2022) for better and faster action sampling, while the DDPM-based diffusion policy (Wang et al., 2022) uses the multi-layer perceptron (MLP) to learn score functions. In contrast to SfBC (Chen et al., 2022), we use the conservative Q-value for policy improvement to learn the score functions, while SfBC only uses the BC loss in the procedure. Moreover, different from DAC, which introduces a score-matching term involving differences between score functions and gradients of the Q-function, our approach integrates value guidance in diffusion-based policy optimization. Unlike MA-DIFF (Zhu et al., 2023) and DoF (Li et al., 2025) that uses an attention-based diffusion model in centralized training and centralized or decentralized execution, our method is decentralized in both the training and execution procedure. Below, we will demonstrate, with extensive experiments, that our DOM2 method

achieves superior performance, significant generalization, and data efficiency compared to the state-of-the-art offline MARL algorithms.

## 5 Experiments

We evaluate our method in different multi-agent environments and datasets. We focus on three primary metrics, performance (how is DOM2 compared to other SOTA baselines), generalization (can DOM2 generalize well if the environment configurations change), and data efficiency (is our algorithm applicable with small datasets and low-quality datasets).

### 5.1 Experiment Setup

**Environments:** We conduct experiments in two widely-used multi-agent tasks including the multi-agent particle environments (MPE) (Lowe et al., 2017) and high-dimensional and challenging multi-agent MuJoCo (MAMuJoCo) tasks (Peng et al., 2021). In MPE, agents known as physical particles need to cooperate with each other to solve the tasks. The MAMuJoCo is an extension for MuJoCo locomotion tasks to enable the robot to run with the cooperation of agents. We use the Predator-prey, World, Cooperative navigation in MPE and 2-agent HalfCheetah in MAMuJoCo as the experimental environments. The details are shown in Appendix A.2.1. To demonstrate the generalization capability of our DOM2 algorithm, we conduct experiments in both standard environments and shifted environments. Compared to the standard environments, the features of the environments are changed randomly to increase the difficulty for the agent to finish the task, which will be shown later.

**Datasets:** We construct 6 different datasets following (Fu et al., 2020) to represent different qualities of behavior policies: random, random-medium, medium-replay, medium, medium-expert and expert dataset. The details are shown in Appendix A.2.3

**Baseline:** We compare the DOM2 algorithm with the following state-of-the-art baseline offline MARL algorithms: MA-CQL (Jiang & Lu, 2021), OMAR (Pan et al., 2022), MA-SfBC as the extension of the single agent diffusion-based policy SfBC (Chen et al., 2022), MA-DIFF (Zhu et al., 2023) and DoF (Li et al., 2025) (Due to dataset mismatch, we compare MA-DIFF and DoF in the ablation study of Section 5.4 and Appendix A.3). Our methods are all built on the independent TD3 with decentralized actors and critics. Each algorithm is executed for 5 random seeds and the mean performance and the standard deviation are presented. A detailed description of hyperparameters, neural network structures, and setup can be found in Appendix A.2.2.

### 5.2 Multi-Agent Particle Environment

**Performace.** Table 1 shows the mean episode returns (same for Table 2 below) of the algorithms under different datasets. We see that in all settings, DOM2 significantly outperforms MA-CQL, OMAR, and MA-SfBC. We also observe that DOM2 has smaller deviations in most settings compared to other algorithms, demonstrating that DOM2 is more stable in different environments.

**Generalization.** In MPE, we design the shifted environment by changing the speed of agents. Specifically, we change the speed of agents by randomly choosing in the region $v_j \in [v_{\min}, 1.0]$ in each episode for evaluation (the default speed of any agent $j$ is all $v_j = 1.0$ in the standard environment). Here $v_{\min} = 0.4, 0.5, 0.3$ in the predator-prey, world, and cooperative navigation, respectively. The values are set to be the minimum speed to guarantee that the agents can all catch the adversary using the slowest speed with an appropriate policy. We train the policy using the dataset generated in the standard environment and evaluate it in the shifted environments to examine the generalization of the policy. The results are shown in the table 2. We can see that DOM2 significantly outperforms the compared algorithms in nearly all settings, and achieves the best performance in 17 out of 18 settings. Only in one setting, the performance is slightly below OMAR.

**Data Efficiency.** In addition to the above performance and generalization, DOM2 also possesses superior data efficiency. To demonstrate this, we train the algorithms using only a small percentage of the samples (fewer full trajectories) in the given dataset (a full dataset contains $10^6$ samples). The results are shown

Table 1: Performance comparison of DOM2 with MA-CQL, OMAR, and MA-SfBC in standard environments of MPE.

| Predator Prey | MA-CQL | OMAR | MA-SfBC | DOM2 |
|---|---|---|---|---|
| Random | 1.0±7.6 | 14.3±9.5 | 3.5±2.5 | **208.7±57.3** |
| Random Medium | 1.7±13.0 | 67.7±30.8 | 12.0±10.7 | **133.0±39.9** |
| Medium Replay | 35.0±21.6 | 86.8±43.7 | 26.1±10.0 | **150.5±23.9** |
| Medium | 101.0±42.5 | 116.9±45.2 | 127.0±50.9 | **155.8±48.1** |
| Medium Expert | 113.2±36.7 | 128.3±35.2 | 152.3±41.2 | **184.4±25.3** |
| Expert | 140.9±33.3 | 202.8±27.1 | 256.0±26.9 | **259.1±22.8** |
| **World** | **MA-CQL** | **OMAR** | **MA-SfBC** | **DOM2** |
| Random | -3.8±3.0 | 0.0±3.3 | -1.8±1.9 | **40.0±14.3** |
| Random Medium | -6.6±1.1 | 28.7±10.4 | 4.0±5.5 | **42.7±9.3** |
| Medium Replay | 15.9±14.2 | 21.1±15.6 | 9.1±5.9 | **65.9±10.6** |
| Medium | 44.3±14.1 | 45.6±16.0 | 54.2±22.7 | **84.5±23.4** |
| Medium Expert | 51.4±25.6 | 71.5±28.2 | 60.6±22.9 | **89.4±16.5** |
| Expert | 57.7±20.5 | 84.8±21.0 | 97.3±19.1 | **99.5±17.1** |
| **Cooperative Navigation** | **MA-CQL** | **OMAR** | **MA-SfBC** | **DOM2** |
| Random | 206.0±17.5 | 211.3±20.3 | 179.8±15.7 | **337.8±26.0** |
| Random Medium | 226.5±22.1 | 272.6±39.4 | 178.8±17.9 | **359.7±28.5** |
| Medium Replay | 229.7±55.9 | 260.7±37.7 | 196.1±11.1 | **324.1±38.6** |
| Medium | 275.4±29.5 | 348.7±51.7 | 276.3±8.8 | **358.9±25.2** |
| Medium Expert | 333.3±50.1 | 450.3±39.0 | 299.8±16.8 | **532.9±54.7** |
| Expert | 478.9±29.1 | 564.6±8.6 | 553.0±41.1 | **628.6±17.2** |

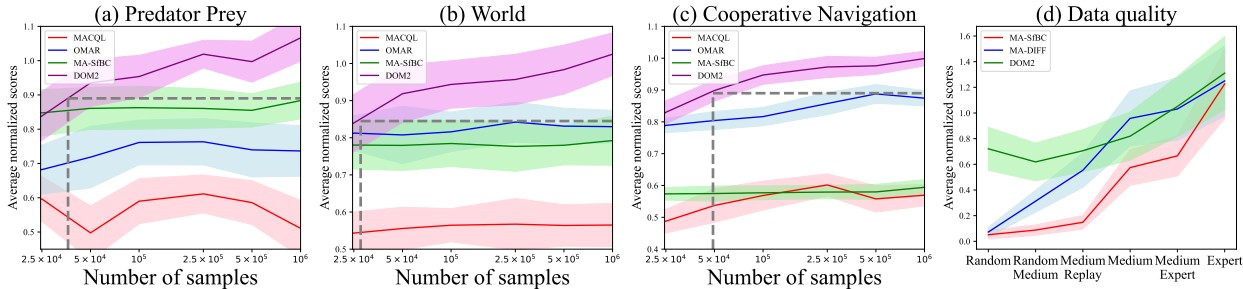

Figure 5: Algorithm performance on data-efficiency. (a)-(c) show the algorithm performance under different numbers of samples. One can see that DOM2 only requires 5% data to achieve the same performance as other baselines. (d) shows the algorithm performance under different data qualities.

in Figure 5 (a)-(c). The averaged normalized score is calculated by averaging the normalized score in 5 different datasets except the medium-replay (the benchmark of the normalized scores is shown in Appendix A.2.1). DOM2 exhibits a remarkably better performance in all MPE tasks, i.e., using a data volume that is 20× times smaller, it still achieves state-of-the-art performance. Moreover, we compare our algorithm with other diffusion-based algorithms, including MA-SfBC (Chen et al., 2022) and MA-DIFF (Zhu et al., 2023) in Figure 5 (d) as the average normalized score among the MPE tasks. DOM2 also significantly outperforms existing algorithms in low-quality dataset, i.e., 10× performance improvement, indicating that DOM2 is highly effective in learning from offline datasets. This unique feature is extremely useful in making good utilization of offline data, especially in applications where data collection can be costly, e.g., robotics and autonomous driving (Chi et al., 2023; Urain et al., 2023).

Table 2: Performance comparison in **shifted environments** of MPE.

| Predator Prey | MA-CQL | OMAR | MA-SfBC | DOM2 |
|---|---|---|---|---|
| Random | 1.8±5.7 | 10.4±3.6 | 9.3±15.4 | **120.7±100.2** |
| Random Medium | 4.0±7.4 | 41.4±20.9 | 22.2±34.8 | **66.2±88.8** |
| Medium Replay | 35.6±24.1 | 60.0±24.9 | 11.9±18.1 | **104.2±132.5** |
| Medium | 80.3±51.0 | 81.1±51.4 | 83.5±97.2 | **95.7±79.9** |
| Medium Expert | 69.5±44.7 | 78.6±59.2 | 84.0±86.6 | **127.9±121.8** |
| Expert | 100.0±37.1 | 151.7±41.3 | 171.6±133.6 | **208.7±160.9** |
| World | MA-CQL | OMAR | MA-SfBC | DOM2 |
| Random | -2.7±3.2 | 1.1±3.4 | -1.9±4.6 | **35.6±23.1** |
| Random Medium | -6.0±7.7 | 28.7±7.4 | 0.0±5.0 | **30.3±34.2** |
| Medium Replay | 8.1±6.2 | 20.1±14.5 | 4.6±9.2 | **51.5±21.3** |
| Medium | 33.3±11.6 | 32.0±15.1 | 35.6±15.4 | **57.5±28.2** |
| Medium Expert | 40.9±15.3 | 44.6±18.5 | 39.3±25.7 | **79.9±39.7** |
| Expert | 51.1±11.0 | 71.1±15.2 | 82.0±33.3 | **91.8±34.9** |
| Cooperative Navigation | MA-CQL | OMAR | MA-SfBC | DOM2 |
| Random | 235.6±19.5 | 251.0±36.8 | 175.5±38.1 | **265.6±57.3** |
| Random Medium | 251.0±36.8 | 266.1±23.6 | 174.3±50.0 | **304.5±45.6** |
| Medium Replay | 224.2±30.2 | 271.3±33.6 | 191.9±54.6 | **302.1±78.2** |
| Medium | 256.5±15.2 | **295.6±46.0** | 285.6±68.2 | 295.2±80.0 |
| Medium Expert | 279.9±21.8 | 373.9±31.8 | 277.9±57.8 | **439.6±89.8** |
| Expert | 376.1±25.2 | 410.6±35.6 | 410.6±83.0 | **444.0±99.0** |

## 5.3 Scalability in Multi-Agent MuJoCo Environment

We now turn to a more complex continuous control task: HalfCheetah-v2 environment in a multi-agent setting (extension of the single-agent task (Peng et al., 2021), detail in Appendix A.2.1).

**Performance.** Table 3 shows the performance of DOM2 in the multi-agent HalfCheetah-v2 environments. We see that DOM2 outperforms other compared algorithms and achieves state-of-the-art performances in all the algorithms and datasets.

**Generalization.** As in the MPE case, we also evaluate the generalization capability of DOM2 in this setting. Specifically, we design shifted environments following the scheme in (Packer et al., 2018), i.e., we set up Random (R) and Extreme (E) environments by changing the environment parameters (details are shown in Appendix A.2.1). The performance of the algorithms is shown in Table 3. The results show that DOM2 significantly outperforms other algorithms in nearly all settings, and achieves the best performance in 11 out of 12 settings.

## 5.4 Ablation study

We conduct an ablation study for DOM2, to evaluate the importance of each component in DOM2 algorithm (diffusion, regularization and data reweighting). Specifically, we compare DOM2 to four modified DOM2 algorithms, each with one different component removed or replaced. The results are shown in Figure 6. We see that removing or replacing any component in DOM2 hurts the performance across all the environments.

In addition to evaluating the components within the DOM2 algorithm, our study includes comparative analyses involving several other algorithms, such as the multi-agent version of BRAC (Wu et al., 2019) (MA-BRAC), IQL (Kostrikov et al., 2021a) (MA-IQL), Diffusion-QL (Wang et al., 2022) (MA-Diffusion-QL) and EDP-DDPM/DPM-Solver (Kang et al., 2023) (MA-EDP-DDPM/DPM-Solver), alongside OMAC (Wang & Zhan, 2023), OMIGA (Wang et al., 2023), CFCQL (Shao et al., 2023), MADIFF (Wang & Zhan, 2023) and

Table 3: Performance comparison of DOM2 with MA-CQL, OMAR, and MA-SfBC in standard and shifted (notated Random and Extreme Env.) MAMuJoCo environments under the HalfCheetah-v2 task.

| HalfCheetah-Standard Env. | MA-CQL | OMAR | MA-SfBC | DOM2 |
|---|---|---|---|---|
| Random | -0.1±0.2 | -0.9±0.1 | -388.9±29.2 | **799.8±143.9** |
| Random Medium | -0.1±0.1 | 219.5±369.1 | -383.1±18.4 | **875.0±155.5** |
| Medium Replay | 1216.6±514.6 | 1674.8±201.5 | -128.3±71.3 | **2564.3±216.9** |
| Medium | 963.4±316.6 | 2797.0±445.7 | 1386.8±248.8 | **2851.2±145.5** |
| Medium Expert | 1989.8±685.6 | 2900.2±403.2 | 1392.3±190.3 | **2919.6±252.8** |
| Expert | 2722.8±1022.6 | 2963.8±410.5 | 2386.6±440.3 | **3676.6±248.1** |
| HalfCheetah-Random Env. | MA-CQL | OMAR | MA-SfBC | DOM2 |
| Random | -0.1±0.3 | -1.0±0.3 | -315.8±25.7 | **581.8±621.0** |
| Random Medium | -0.2±0.3 | 90.8±176.2 | -327.0±21.0 | **1245.8±315.9** |
| Medium Replay | 1279.6±305.4 | 1648.0±132.6 | -171.4±43.7 | **2290.8±128.5** |
| Medium | 1111.7±585.9 | 2650.0±201.5 | 1367.6±203.9 | **2788.5±112.9** |
| Medium Expert | 1291.5±408.3 | 2616.6±368.8 | 1442.1±218.9 | **2731.7±268.1** |
| Expert | 2678.2±900.9 | 2295.0±357.2 | 2397.4±670.3 | **3178.7±370.5** |
| HalfCheetah-Extreme Env. | MA-CQL | OMAR | MA-SfBC | DOM2 |
| Random | -0.1±0.1 | -1.0±0.3 | -309.8±23.0 | **372.9±449.7** |
| Random Medium | -0.1±0.2 | 129.8±374.6 | -329.2±43.6 | **482.0±468.6** |
| Medium Replay | 1290.4±230.8 | 1549.9±311.4 | -169.8±50.5 | **1904.2±201.8** |
| Medium | 1108.1±944.0 | 2197.4±95.2 | 1355.0±195.7 | **2232.4±215.1** |
| Medium Expert | 1127.1±565.2 | 2196.9±186.9 | 1393.7±347.7 | **2219.0±170.7** |
| Expert | 2117.0±524.0 | 1615.7±707.6 | **2757.2±200.6** | 2641.3±382.9 |

Table 4: Comparison among algorithms in the Predator Prey task.

| Algorithm | Random | Medium Replay | Medium | Expert |
|---|---|---|---|---|
| MA-BRAC | 29.9±29.3 | 45.2±18.1 | 44.5±19.6 | 38.5±11.4 |
| MA-IQL | 3.0±5.5 | 37.0±38.1 | 105.6±48.2 | 219.7±27.4 |
| MA-Diffusion-QL | 82.2±22.6 | 83.9±18.4 | 117.1±45.0 | 224.6±29.5 |
| MA-EDP-DDPM | 8.8±8.5 | 70.0±47.0 | 111.9±46.6 | 217.8±20.8 |
| MA-EDP-DPM-Solver | 64.3±14.3 | 69.4±8.9 | 109.0±39.7 | 207.9±15.4 |
| OMAC | 19.8±17.5 | 20.5±21.5 | 48.1±25.7 | 72.6±56.7 |
| OMIGA | -2.0±7.5 | 8.4±15.7 | 28.3±24.4 | 62.0±41.8 |
| CFCQL | 144.8±29.6 | 130.8±11.4 | 125.8±41.4 | 220.1±24.9 |
| MADIFF | 2.0±7.6 | 114.1±17.5 | 142.3±19.7 | 225.2±27.7 |
| DoF | 24.0±6.1 | 94.0±19.2 | 155.1±18.2 | 223.7±12.0 |
| DOM2 | **208.7±57.3** | **150.5±23.9** | **155.8±48.1** | **259.1±22.8** |

DoF (Li et al., 2025) (offline MARL algorithms with centralized training and decentralized execution). The assessment, conducted across the Predator-Prey task using four distinct datasets, unequivocally demonstrates that DOM2 exhibits superior performance compared to these algorithms, establishing its state-of-the-art

capabilities. This outcome strongly substantiates the inherent advantages encapsulated within the DOM2 algorithm.

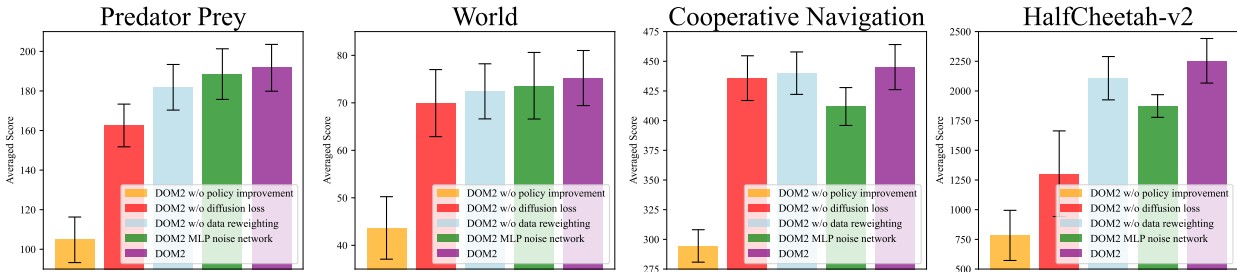

Figure 6: Impact of different algorithm components. We compare DOM2 (purple) with DOM2 w/o policy improvement (orange), DOM2 w/o diffusion loss (red), DOM2 w/o data reweighting (lightblue) and DOM2 using a MLP-based (Multi-Layer Perceptron) noise network in diffusion (green). The results show that every component of DOM2 contributes to its performance improvement.

We also investigate the sensitivity to key hyperparameters: the regularization coefficient $\eta$ and the diffusion step $N$.

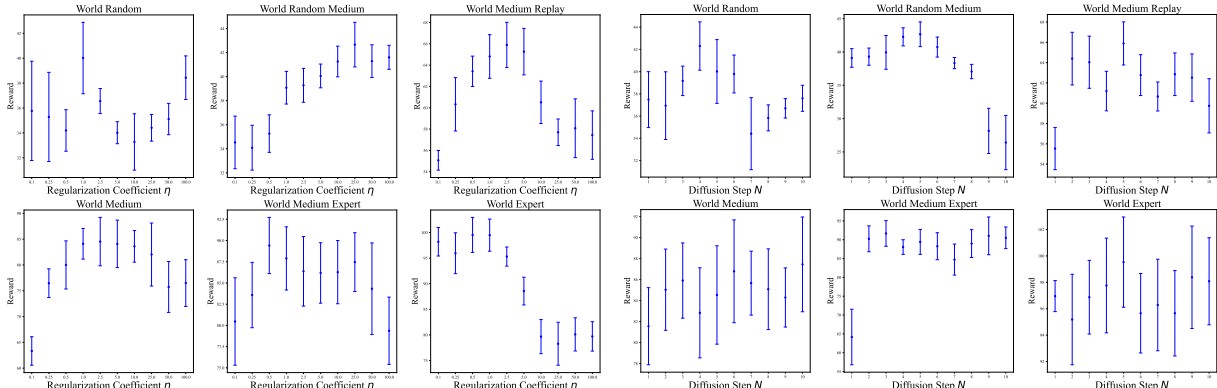

Figure 7: Left: The effect of the $\eta$ value in MPE World in 6 different datasets. Right: The effect of the diffusion step $N$ in MPE World in 6 different datasets.

**The effect of the regularization coefficient** $\eta$    Figure 7a shows the averaged mean episode returns of DOM2 over the MPE world task with different values of the regularization coefficient $\eta \in [0.1, 100.0]$ in 6 datasets. In order to perform the advantage of the diffusion-based policy, the appropriate coefficient value $\eta$ needs to balance the two regularization terms appropriately, which is influenced by the performance of the dataset. For the expert dataset, $\eta$ tends to be small, and in other datasets, $\eta$ tends to be relatively larger. The reason that small $\eta$ performs well in the expert dataset is that with data from well-trained strategies, getting close to the behavior policy is sufficient for training a policy without policy improvement.

**The effect of the diffusion step** $N$    Figure 7b shows the averaged mean episode returns of DOM2 over the MPE world task with different values of the diffusion step $N \in [1, 10]$ under each dataset. The numbers of optimal diffusion steps vary with the dataset. We also observe that $N = 5$ is a good choice for both efficiency of diffusion-based action generation and the performance of the obtained policy in MPE.

# 6    Conclusion

We propose DOM2, a novel offline MARL algorithm, which contains three key components, i.e., a diffusion mechanism for enhancing policy expressiveness and diversity, an appropriate regularizer, and a data-

reweighting method. Through extensive experiments on multi-agent particle and multi-agent MuJoCo environments, we show that DOM2 significantly outperforms state-of-the-art benchmarks. Moreover, DOM2 possesses superior generalization capability and ultra-high data efficiency, i.e., achieving the same performance as benchmarks with 20+ times less data.

## 7 Acknowledgement

This work was supported by the National Natural Science Foundation of China Grant 52494974 and the Tsinghua University - Keystone Electrical (Zhejiang) Co.,Ltd Joint Research Center for Embodied Multimodal Artificial Intelligence (JCEMAI).

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

# A    Appendix

## A.1    Additional Details about Diffusion Probabilistic Model

In this section, we elaborate on more details about the diffusion probabilistic model that we do not cover in Section 4.1 due to space limitation, and compare the similar parts between the diffusion model and DOM2 in offline MARL.

In the noising action part, we emphasize a forward process $\{b_{t,j}^{\tau}\}_{\tau \in [\tau_0, \tau_N]}$ starting at $b_{t,j}^{\tau_0} \sim \pi_{\theta_j}(\cdot|o_{t,j})$ in the dataset $\mathcal{D}$ and $b_{t,j}^{\tau_N}$ is the final noise. This forward process satisfies that for any diffusing time index $\tau \in [\tau_0, \tau_N]$, the transition probability $q_{\tau_0\tau}(b_{t,j}^{\tau}|b_{t,j}^{\tau_0}) = \mathcal{N}(b_{t,j}^{\tau}; \alpha_\tau b_{t,j}^{\tau_0}, \sigma_\tau^2 I)$ (Lu et al., 2022) ($\alpha_\tau, \sigma_\tau$ is called the noise schedule). We build the reverse process of SDE as equation 2 and we will describe the connection between the forward process and the reverse process of SDE. Kingma (Kingma et al., 2021) proves that the following forward SDE (equation 6) solves to a process whose transition probability $q_{\tau_0\tau}(b_{t,j}^{\tau}|b_{t,j}^{\tau_0})$ is the same as the forward process, which is written as:

$$d b_{t,j}^{\tau} = f(\tau) b_{t,j}^{\tau} d\tau + g(\tau) d w_\tau, \quad b_{t,j}^{\tau_0} \sim \pi_{\beta_j}(\cdot|o_{t,j}). \tag{6}$$

Here $\pi_{\beta_j}(\cdot|o_{t,j})$ is the behavior policy to generate $b_{t,j}^{\tau_0}$ for agent $j$ given the observation $o_{t,j}$, $f(\tau) = \frac{d\log\alpha_\tau}{d\tau}, g^2(\tau) = \frac{d\sigma_\tau^2}{d\tau} - 2\frac{d\log\alpha_\tau}{d\tau}\sigma_\tau^2$ and $w_t$ is a Brownian motion. It was proven in (Song et al., 2020b) that the forward process of SDE from $\tau_0$ to $\tau_N$ has an equivalent reverse process of the SDE from $\tau_N$ to $\tau_0$, which is the equation 2. In this way, the forward process of conditional probability and the reverse process of SDE are connected.

In our DOM2 for offline MARL, we propose the objective function in equation 3 and its simplification. In detail, following (Lu et al., 2022), the loss function for score matching is defined as:

$$
\begin{aligned}
\mathcal{L}_{bc}(\theta_j) &:= \int_{\tau_0}^{\tau_N} \omega(\tau) \mathbb{E}_{a_{t,j}^{\tau} \sim q_\tau(b_{t,j}^{\tau})}[\|\epsilon_{\theta_j}(a_{t,j}^{\tau}, o_{t,j}, \tau) + \sigma_\tau \nabla_{b_{t,j}^{\tau}} \log q_\tau(b_{t,j}^{\tau}|o_{t,j})\|_2^2] d\tau \\
&= \int_{\tau_0}^{\tau_N} \omega(\tau) \mathbb{E}_{a_{t,j}^{\tau_0} \sim \pi_{\beta_j}(a_{t,j}^{\tau_0}|o_{t,j}), \epsilon \sim \mathcal{N}(0, I)}[\|\epsilon - \epsilon_{\theta_j}(\alpha_\tau a_{t,j}^{\tau_0} + \sigma_\tau \epsilon, o_{t,j}, \tau)\|_2^2] d\tau + C,
\end{aligned}
\tag{7}
$$

where $\omega(\tau)$ is the weighted parameter and $C$ is a constant independent of $\theta_j$. In practice for simplification, we set that $w(\tau) = 1/(\tau_N - \tau_0)$, replace the integration by random sampling a diffusion timestep and ignore the equally weighted parameter $\omega(\tau)$ and the constant $C$. After these simplifications, the final objective becomes equation 3.

Next, we introduce the accelerated sampling method to build the connection between the reverse process of SDE for sampling and the accelerated DPM-solver.

In the denoising part, we utilize the following SDE of the reverse process (Equation (2.5) in (Lu et al., 2022)) as:

$$d a_{t,j}^{\tau} = \left[ f(\tau) a_{t,j}^{\tau} + \frac{g^2(\tau)}{\sigma_\tau} \epsilon_{\theta_j}(a_{t,j}^{\tau}, o_{t,j}, \tau) \right] d\tau + g(\tau) d\overline{w}_\tau, \tag{8}$$

where $a_{t,j}^{\tau_N} \sim \mathcal{N}(0, I)$. To achieve faster sampling, (Song et al., 2020b) proves that the following ODE equivalently describes the process given by the reverse diffusion SDE. It is thus called the diffusion ODE.

$$\frac{d a_{t,j}^{\tau}}{d\tau} = f(\tau) a_{t,j}^{\tau} + \frac{g^2(\tau)}{2\sigma_\tau} \epsilon_{\theta_j}(a_{t,j}^{\tau}, o_{t,j}, \tau), \quad a_{t,j}^{\tau_N} \sim \mathcal{N}(0, I). \tag{9}$$

At the end of the denoising part, we use the efficient DPM-solver (equation 4) to solve the diffusion ODE and thus implement the denoising process. The formal derivation can be found on (Lu et al., 2022) and we restate their argument here for the sake of completeness, for a more detailed explanation, please refer to (Lu et al., 2022).

For such a semi-linear structured ODE in equation 9, the solution at time $\tau$ can be formulated as:

$$\boldsymbol{a}_{t,j}^{\tau} = \exp\left(\int_{\tau'}^{\tau} f(u)\mathrm{d}u\right)\boldsymbol{a}_{t,j}^{\tau'} + \int_{\tau'}^{\tau}\left(\exp\left(\int_{u}^{\tau} f(z)\mathrm{d}z\right)\frac{g^2(u)}{2\sigma_u}\boldsymbol{\epsilon}_{\boldsymbol{\theta}_j}(\boldsymbol{a}_{t,j}^{u},\boldsymbol{o}_{t,j},u)\right)\mathrm{d}u. \tag{10}$$

Defining $\lambda_\tau = \log(\alpha_\tau/\sigma_\tau)$, we can rewrite the solution as:

$$\boldsymbol{a}_{t,j}^{\tau} = \frac{\alpha_\tau}{\alpha_\tau'}\boldsymbol{a}_{t,j}^{\tau'} - \alpha_\tau\int_{\tau'}^{\tau}\left(\frac{\mathrm{d}\lambda_u}{\mathrm{d}u}\right)\frac{\sigma_u}{\alpha_u}\boldsymbol{\epsilon}_{\boldsymbol{\theta}_j}(\boldsymbol{a}_{t,j}^{u},\boldsymbol{o}_{t,j},u)\mathrm{d}u. \tag{11}$$

Notice that the definition of $\lambda_\tau$ is dependent on the noise schedule $\alpha_\tau, \sigma_\tau$. If $\lambda_\tau$ is a continuous and strictly decreasing function of $\tau$ (the selection of our final noise schedule in equation 13 actually satisfies this requirement, which we will discuss afterwards), we can rewrite the term by *change-of-variable*. Based on the inverse function $\tau_\lambda(\cdot)$ from $\lambda$ to $\tau$ such that $\tau = \tau_\lambda(\lambda_\tau)$ (for simplicity we can also write this term as $\tau_\lambda$) and define $\hat{\boldsymbol{\epsilon}}_{\boldsymbol{\theta}_j}(\hat{\boldsymbol{a}}_{t,j}^{\lambda_\tau},\boldsymbol{o}_{t,j},\lambda_\tau) = \boldsymbol{\epsilon}_{\boldsymbol{\theta}_j}(\boldsymbol{a}_{t,j}^{\tau},\boldsymbol{o}_{t,j},\tau)$, we can rewrite equation 11 as:

$$\boldsymbol{a}_{t,j}^{\tau} = \frac{\alpha_\tau}{\alpha_\tau'}\boldsymbol{a}_{t,j}^{\tau'} - \alpha_\tau\int_{\lambda_{\tau'}}^{\lambda_\tau}\exp\left(-\lambda\right)\hat{\boldsymbol{\epsilon}}_{\boldsymbol{\theta}_j}(\hat{\boldsymbol{a}}_{t,j}^{\lambda},\boldsymbol{o}_{t,j},\lambda)\mathrm{d}\lambda. \tag{12}$$

equation 12 is satisfied for any $\tau, \tau' \in [\tau_0, \tau_N]$. We uniformly partition the diffusion horizon $[\tau_0, \tau_N]$ into $N$ subintervals $\{[\tau_i, \tau_{i+1}]\}_{i=0}^{N-1}$, where $\tau_i = i/N$ (also $\tau_0 = 0, \tau_N = 1$). We follow (Xiao et al., 2021) to use the variance-preserving (VP) type function (Ho et al., 2020; Song et al., 2020b; Lu et al., 2022) to train the policy efficiently. First, define $\{\beta_\tau\}_{\tau\in[0,1]}$ by

$$\beta_\tau = 1 - \exp\left(-\beta_{\min}\frac{1}{(N+1)} - (\beta_{\max} - \beta_{\min})\frac{2N\tau+1}{2(N+1)^2}\right), \tag{13}$$

and we pick $\beta_{\min} = 0.1, \beta_{\max} = 20.0$. Then we choose the noise schedule $\alpha_{\tau_i}, \sigma_{\tau_i}$ by $\alpha_{\tau_i} = 1 - \beta_{\tau_i}, \sigma_{\tau_i}^2 = 1 - \alpha_{\tau_i}^2$ for $i = 1\ldots N$. It can be then verified that by plugging this particular choice of $\alpha_\tau$ and $\sigma_\tau$ into $\lambda_\tau = \log(\alpha_\tau/\sigma_\tau)$, the obtained $\lambda_\tau$ is a strictly decreasing function of $\tau$ (Appendix E in (Lu et al., 2022)).

In each interval $[\tau_{i-1}, \tau_i]$, given $\boldsymbol{a}_{t,j}^{\tau_i}$, the action obtained in the previous diffusion step at $\tau_i$, according to equation 12, the exact action in the next step denoted as $\boldsymbol{a}_{t,j}^{\tau_{i-1}}$ is given by:

$$\boldsymbol{a}_{t,j}^{\tau_{i-1}} = \frac{\alpha_{\tau_{i-1}}}{\alpha_{\tau_i}}\boldsymbol{a}_{t,j}^{\tau_i} - \alpha_{\tau_i}\int_{\lambda_{\tau_i}}^{\lambda_{\tau_{i-1}}}\exp\left(-\lambda\right)\hat{\boldsymbol{\epsilon}}_{\boldsymbol{\theta}_j}(\hat{\boldsymbol{a}}_{t,j}^{\lambda},\boldsymbol{o}_{t,j},\lambda)\mathrm{d}\lambda. \tag{14}$$

We take the $k$-th order Taylor expansion for $\hat{\boldsymbol{\epsilon}}_{\boldsymbol{\theta}_j}(\hat{\boldsymbol{a}}_{t,j}^{\lambda},\boldsymbol{o}_{t,j},\lambda)$ at $\lambda_{\tau_i}$ and denote the derivative of $\hat{\boldsymbol{\epsilon}}_{\boldsymbol{\theta}_j}(\hat{\boldsymbol{a}}_{t,j}^{\lambda},\boldsymbol{o}_{t,j},\lambda)$ in the $k$-th order as $\hat{\boldsymbol{\epsilon}}_{\boldsymbol{\theta}_j}^{(k)}(\hat{\boldsymbol{a}}_{t,j}^{\lambda},\boldsymbol{o}_{t,j},\lambda_{\tau_i})$. By ignoring the higher-order remainder $\mathcal{O}((\lambda_{\tau_{i-1}} - \lambda_{\tau_i})^{k+1})$, the $k$-th order DPM-solver for sampling can be written as:

$$\boldsymbol{a}_{t,j}^{\tau_{i-1}} = \frac{\alpha_{\tau_{i-1}}}{\alpha_{\tau_i}}\boldsymbol{a}_{t,j}^{\tau_i} - \alpha_{\tau_i}\sum_{n=0}^{k-1}\hat{\boldsymbol{\epsilon}}_{\boldsymbol{\theta}_j}^{(n)}(\hat{\boldsymbol{a}}_{t,j}^{\lambda_{\tau_i}},\boldsymbol{o}_{t,j},\lambda_{\tau_i})\int_{\lambda_{\tau_i}}^{\lambda_{\tau_{i-1}}}\exp\left(-\lambda\right)\frac{(\lambda-\lambda_{\tau_i})^n}{n!}\mathrm{d}\lambda. \tag{15}$$

For $k = 1$, the results are actually the first-order iteration function in Section 4.1. Similarly, we can use a higher-order DPM-solver.

## A.2 Experimental Details

### A.2.1 Experimental Setup: Environments

We implement our algorithm and baselines based on the open-source environmental engines of multi-agent particle environments (MPE) (Lowe et al., 2017),[1] and multi-agent MuJoCo environments (MAMu-JoCo) (Peng et al., 2021)[2]. Figure 8 shows the tasks in MPE and MAMuJoCo. In cooperative navigation

---

[1]https://github.com/openai/multiagent-particle-envs
[2]https://github.com/schroederdewitt/multiagent_mujoco

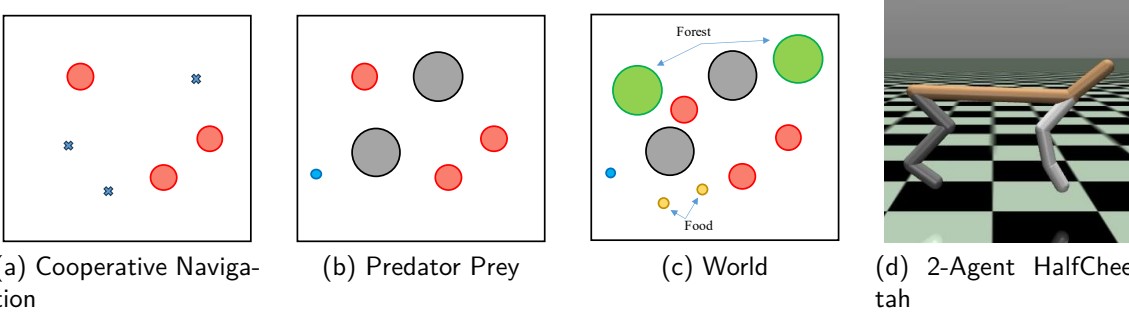

| (a) Cooperative Naviga-tion | (b) Predator Prey | (c) World | (d) 2-Agent HalfChee-tah |

Figure 8: Multi-agent particle environments (MPE) and Multi-agent HalfCheetah task in MuJoCo Environment (MAMuJoCo).

shown in Figure 8a, agents (red dots) cooperate to reach the landmark (blue crosses) without collision. In predator-prey in Figure 8b, predators (red dots) are intended to catch the prey (blue dots) and avoid collision with the landmark (grey dots). The predators need to cooperate with each other to surround and catch the prey because the predators run slower than the prey. The world task in Figure 8c consists of 3 agents (red dots) and 1 adversary (blue dots). The slower agents are intended to catch the faster adversary that desires to eat food (yellow dots). The agents need to avoid collision with the landmark (grey dots). Moreover, if the adversary hides in the forest (green dots), it is harder for the agents to catch the adversary because they do not know the position of the adversary. The two-agent HalfCheetah is shown in Figure 8d, and different agents control different joints (grey and white joints) and they need to cooperate for better control the half-shaped cheetah to run stably and fast. The expert and random scores (a.k.a., mean episode returns) for cooperative navigation, predator-prey, and world are $\{516.8, 159.8\}, \{185.6, -4.1\}, \{79.5, -6.8\}$, and we use these scores to calculate the normalized scores in Figure 5.

For the MAMuJoCo environment, we design two different shifted environments: Random (R) environment and Extreme (E) environments following (Packer et al., 2018). These environments have different parameters and we focus on randomly sampling the three parameters: (1) power, the parameter to influence the force that is multiplied before application, (2) torso density, the parameter to influence the weight, (3) sliding friction of the joints. The detailed sample regions of these parameters in different environments are shown in Table 5.

Table 5: Range of parameters in the MAMuJoCo HalfCheetah-v2 environment.

|          | Deterministic | Random      | Extreme                     |
|----------|---------------|-------------|-----------------------------|
| Power    | 1.0           | [0.8,1.2]   | [0.6,0.8]∪[1.2,1.4]         |
| Density  | 1000          | [750,1250]  | [500,750]∪[1250,1500]       |
| Friction | 0.4           | [0.25,0.55] | [0.1,0.25]∪[0.55,0.7]       |

### A.2.2 Experimental Setup: Network Structures and Hyperparameters

In DOM2, we utilize the multi-layer perceptron (MLP) to model the Q-value functions of the critics by concatenating the state-action pairs and sending them into the MLP to generate the Q-function, which is the same as in MA-CQL and OMAR (Pan et al., 2022). Different from MA-CQL and OMAR that uses MLP for action generation, we utilize the diffusion policy to generate actions. We use a multi-layer residual network to model the noise network $\epsilon_{\boldsymbol{\theta}_j}(\boldsymbol{a}_{t,j}^{\tau_i}, \boldsymbol{o}_{t,j}, \tau_i)$ for agent $j$ at timestep $\tau_i$, which ensembles the U-Net architecture (Chen et al., 2022; Janner et al., 2022). One difference is that we use a dropout layer with a 0.1 dropout rate in each residual network component for preventing overfitting and better training stability.

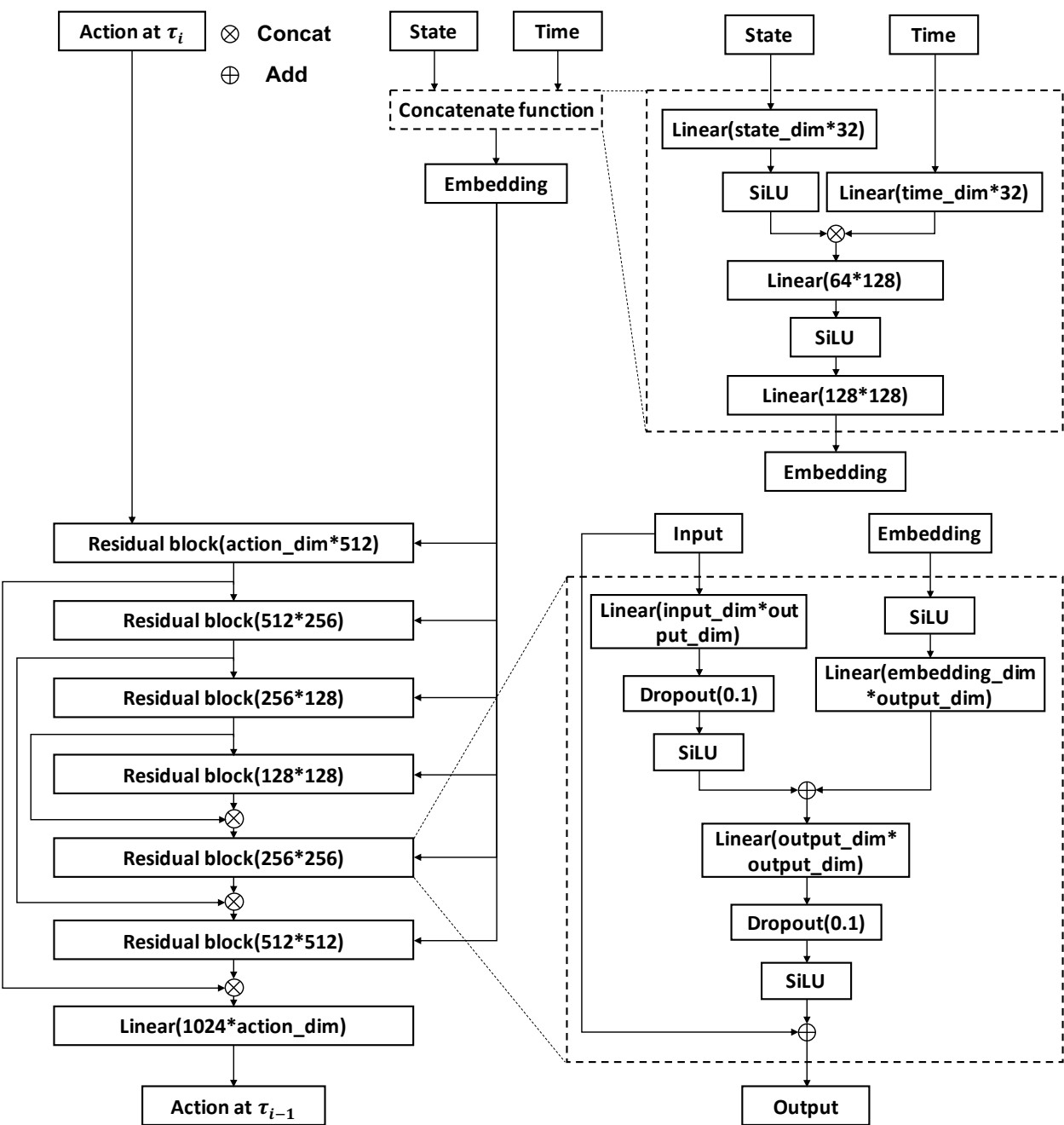

Figure 9: Architecture of the noise network $\epsilon_{\theta_j}$ as is a multi-layer residual network that resembles the structure of U-Net (Ho et al., 2020; Chen et al., 2022). Different from (Chen et al., 2022), we include a dropout layer for training stability.

All the MLPs consist of 1 batch normalization layer, 2 hidden layers, and 1 output layer with the size (input_dim, hidden_dim), (hidden_dim, hidden_dim), (hidden_dim, output_dim) and hidden_dim = 256. In the hidden layers, the output is activated with the Mish function, and the output of the output layer is activated with the Tanh function[3].

---

[3]Code is available at: https://github.com/lizr16/DOM2

For training the Q-value network, we use the learning rate of $3 \times 10^{-4}$ in all environments. In policy training, we use $5 \times 10^{-3}$ in all MPE environments as the learning rate to train the noise network (Figure 9) in the diffusion policy. In the MAMuJoCo HalfCheetah-v2 environment, the learning rates for training the noise network in random, random-medium, medium-replay, medium, medium-expert, and expert datasets are set to $1 \times 10^{-3}, 2.5 \times 10^{-4}, 1 \times 10^{-4}, 2.5 \times 10^{-4}, 2.5 \times 10^{-4}, 5 \times 10^{-4}$, respectively. The total diffusion step number $N$ is for sampling denoised actions. We use $N = 5$ as the diffusion timestep in MPE and $N = 10$ in the MAMuJoCo HalfCheetah-v2 environment. The trade-off parameter $\eta$ is used to balance the regularizers of actor losses and the threshold values $\mathcal{R} = \{r_{\text{th},1}, ..., r_{\text{th},K}\}$ are set for efficient data reweighting. The hyperparameter $\eta$ and the set of threshold values $\mathcal{R} = \{r_{\text{th},1}, ..., r_{\text{th},K}\}$ in different settings are shown in Table 6. For all other hyperparameters, we use the same values in our experiments.

Table 6: The $\eta$ value and the set of threshold values $\mathcal{R}$ in DOM2.

| Predator Prey | $\eta$ | Set of threshold values $\mathcal{R}$ |
|---|---|---|
| Random | 25.0 | None |
| Random Medium | 250.0 | $[100.0, 150.0, 200.0, 250.0, 300.0]$ |
| Medium Replay | 5.0 | $[0.0, 10.0, 20.0, 30.0, 40.0, 50.0, 60.0, 70.0, 80.0, 90.0, 100.0]$ |
| Medium | 2.5 | $[100.0, 150.0, 200.0, 250.0, 300.0]$ |
| Medium Expert | 25.0 | $[100.0, 150.0, 200.0, 250.0, 300.0, 350.0, 400.0]$ |
| Expert | 0.5 | $[200.0, 250.0, 300.0, 350.0, 400.0]$ |
| **World** | $\eta$ | Set of threshold values $\mathcal{R}$ |
| Random | 5.0 | None |
| Random Medium | 25.0 | $[65.5, 86.4, 101.5, 101.5]$ |
| Medium Replay | 2.5 | $[-3.7, 5.9, 15.6, 15.6]$ |
| Medium | 0.5 | $[65.5, 86.4, 101.5, 101.5]$ |
| Medium Expert | 0.5 | $[50.0, 75.0, 100.0, 125.0, 150.0, 175.0]$ |
| Expert | 0.5 | $[75.0, 100.0, 125.0, 150.0, 175.0]$ |
| **Cooperative Navigation** | $\eta$ | Set of threshold values $\mathcal{R}$ |
| Random | 10000.0 | None |
| Random Medium | 500.0 | $[200.0, 250.0, 300.0, 350.0, 400.0, 450.0, 500.0, 550.0]$ |
| Medium Replay | 500.0 | $[0.0, 10.0, 20.0, 30.0, 40.0, 50.0, 60.0, 70.0, 80.0, 90.0, 100.0]$ |
| Medium | 250.0 | $[200.0, 250.0, 300.0, 350.0, 400.0, 450.0, 500.0, 550.0]$ |
| Medium Expert | 250.0 | $[264.4, 267.3, 333.5, 336.4, 385.3, 385.3, 387.9, 387.9]$ |
| Expert | 50.0 | $[525.0, 550.0, 575.0, 600.0, 625.0]$ |
| **HalfCheetah** | $\eta$ | Set of threshold values $\mathcal{R}$ |
| Random | 0.5 | None |
| Random Medium | 0.5 | $[1800.0, 1850.0, 1900.0, 1950.0, 2000.0]$ |
| Medium Replay | 1.0 | $[100.0, 300.0, 500.0, 1000.0, 1500.0]$ |
| Medium | 2.5 | $[1800.0, 1850.0, 1900.0, 1950.0, 2000.0]$ |
| Medium Expert | 2.5 | $[1631.6, 1692.5, 1735.5, 1735.5]$ |
| Expert | 0.05 | $[3800.0, 3850.0, 3900.0, 3950.0, 4000.0]$ |

### A.2.3 Experimental Setup: Dataset construction

We construct 6 different datasets following (Fu et al., 2020) to represent different qualities of behavior policies: i) Random dataset: take 1 million samples by unrolling a randomly initialized policy, ii) Medium-replay dataset: record all of the samples in the replay buffer during training until the performance of the policy is at the medium level, iii) Medium dataset: take 1 million samples by unrolling a policy whose performance reaches the medium level, vi) Expert dataset: take 1 million samples by unrolling a well-trained policy, v) Random-medium dataset: take 1 million samples by sampling the random dataset and the medium

dataset in proportion (90% random dataset and 10% medium dataset in MPE, 99.9% random dataset and 0.1% medium dataset in MAMuJoCo). and vi) Medium-expert dataset: take 1 million samples by sampling the medium dataset and the expert dataset in proportion (90% medium dataset and 10% expert dataset in MPE, 99.9% medium dataset and 0.1% expert dataset in MAMuJoCo).

### A.2.4   Details about 3-Agent 6-Landmark Task

We now discuss detailed results in the 3-Agent 6-Landmark task. We construct the environment based on the cooperative navigation task in multi-agent particles environment (Lowe et al., 2017). This task contains 3 agents and 6 landmarks. The size of agents and landmarks are all 0.1. For any landmark $j = 0, 1, ..., 5$, its position is given by $(\cos(\frac{2\pi j}{6}), \sin(\frac{2\pi j}{6}))$. In each episode, the environment initializes the positions of 3 agents inside the circle of the center $(0, 0)$ with a 0.1 radius uniformly at random. If the agent can successfully find any one of the landmarks, the agent gains a positive reward. If two agents collide, the agents are both penalized with a negative reward.

We construct two different environments: the standard environment and shifted environment. In the standard environment, all 6 landmarks exist in the environment, while in the shifted environment, in each episode, we randomly hide 3 out of 6 landmarks. We collect data generated from the standard environment and train the agents using different algorithms for both environments.

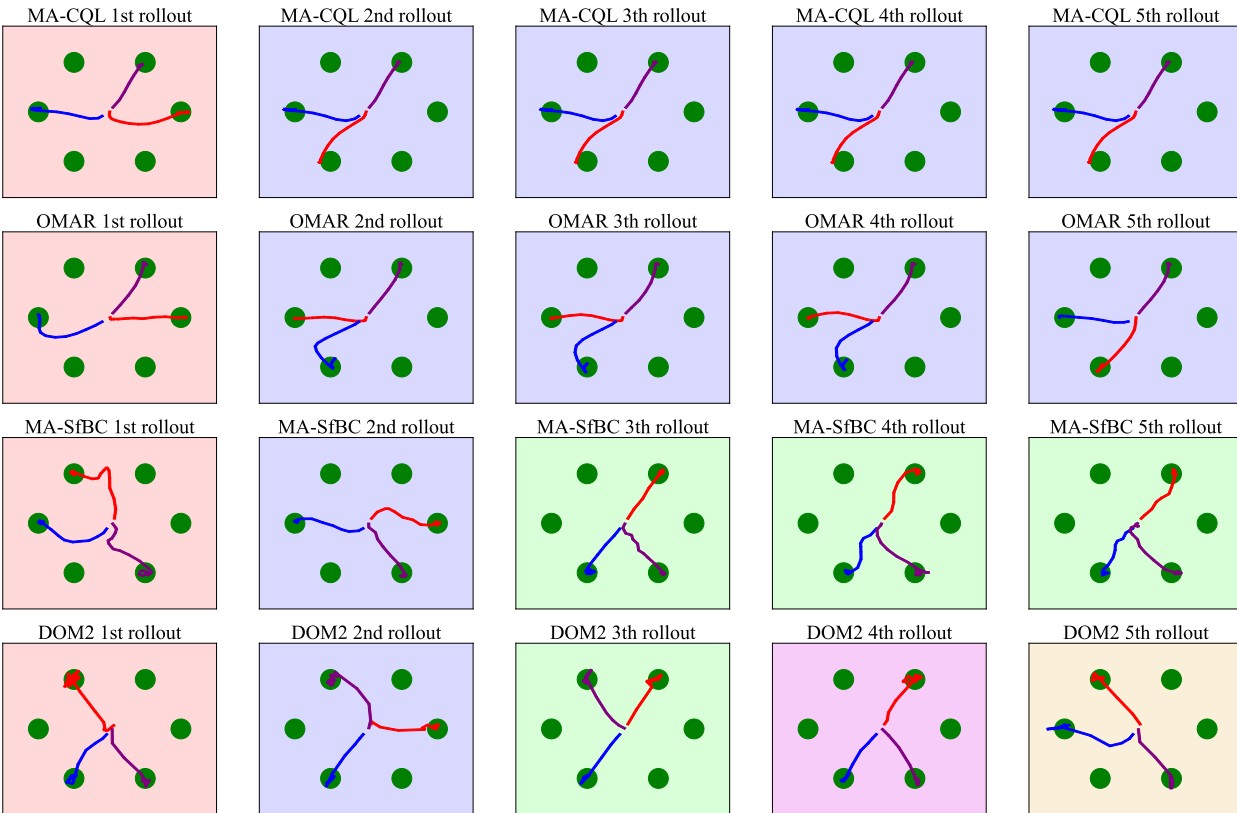

Figure 10: Visualization of trajectories generated by different algorithms with the same initialized position (the center of space) for the 3-agent 6-landmark setting. The red, blue and purple lines are trajectories of the three agents, and all landmarks are colored green. We use different background colors to denote different strategies (i.e., ways to reach three landmarks and complete the tasks) found by the algorithms. We notice that DOM2 finds 5 strategies, while the numbers of strategies found by MA-CQL, OMAR and MA-SfBC are 2, 2 and 3, respectively.

We evaluate how our algorithm performs compared to the baseline algorithms in this task (with different configurations of the targets) and investigate their performance by rolling out $K$ times at each evaluation ($K \in \{1, 10\}$) following (Kumar et al., 2020b). For evaluating the policy in the standard environment, we test the policy for 10 episodes with different initialized positions and calculate the mean value (a.k.a, mean episode returns, same below) and the standard deviation as the results of evaluating the policy. This corresponds to rolling out $K = 1$ time at each evaluation. For the shifted environment, in spite of the former evaluation method ($K = 1$), we also evaluate the algorithm in another way following (Kumar et al., 2020b). We first test the policy for 10 episodes at the same initialized positions and take the maximum return in these 10 episodes. We repeat this procedure 10 times with different initialized positions and calculate the mean value and the standard deviation as the results of evaluating the policy, which corresponds to rolling out $K = 10$ times at each evaluation. It has been reported (see (Kumar et al., 2020b)) that for a diversity-driven method, increasing $K$ can help the diverse policy gain higher returns.

In Table 7, we show the results (the mean episode returns) of different algorithms in standard environments and shifted environments. It can be seen that DOM2 outperforms other algorithms in both the standard environment and shifted environments. Specifically, in the standard environment, DOM2 outperforms other algorithms. This shows that DOM2 has better expressiveness compared to other algorithms. In the shifted environment, when $K = 1$, it turns out that DOM2 already achieves better performance with expressiveness. Moreover, when $K = 10$, DOM2 significantly improves the performance compared to the results in the $K = 1$ setting. This implies that DOM2 finds much more diverse policies, thus achieving better performance compared to the existing conservatism-based method, i.e., MA-CQL and OMAR.

Table 7: Comparison of DOM2 with other algorithms in 3-Agent 6-Landmark settings under the standard environment and $K = 1$, the shifted environment and $K = 1$, and the shifted environment and $K = 10$ in policy evaluation.

| Standard Environment $K = 1$ | MA-CQL | OMAR | MA-SfBC | DOM2 |
|---|---|---|---|---|
| Random | 321.2±39.1 | 326.0±39.4 | 198.5±23.5 | **470.0±70.0** |
| Random Medium | 237.4±48.2 | 237.8±55.9 | 201.5±19.0 | **329.9±60.0** |
| Medium Replay | 396.9±40.1 | 455.7±52.5 | 339.3±29.5 | **542.4±32.5** |
| Medium | 267.4±37.2 | 349.9±20.7 | 459.9±25.2 | **532.5±55.2** |
| Medium Expert | 300.9±77.4 | 395.5±91.0 | 552.1±16.9 | **678.7±4.4** |
| Expert | 457.5±110.0 | 595.0±54.7 | 606.1±13.9 | **683.3±2.1** |
| Shifted Environment $K = 1$ | MA-CQL | OMAR | MA-SfBC | DOM2 |
| Random | 177.5±24.4 | 178.4±34.3 | 142.2±13.0 | **262.5±42.7** |
| Random Medium | 157.0±33.4 | 153.3±31.5 | 147.2±13.4 | **196.2±31.8** |
| Medium Replay | 247.0±43.4 | 274.4±18.0 | 205.7±37.5 | **317.2±54.7** |
| Medium | 171.6±21.8 | 214.0±18.0 | 276.7±48.9 | **284.8±37.6** |
| Medium Expert | 201.2±54.7 | 241.9±32.2 | 328.7±45.9 | **382.3±36.4** |
| Expert | 258.1±67.5 | 334.0±21.7 | 374.2±28.5 | **393.1±43.3** |
| Shifted Environment $K = 10$ | MA-CQL | OMAR | MA-SfBC | DOM2 |
| Random | 186.8±13.9 | 186.7±30.1 | 203.5±12.2 | **283.7±53.0** |
| Random Medium | 160.8±31.4 | 164.5±37.7 | 206.9±9.9 | **217.3±30.6** |
| Medium Replay | 253.3±39.3 | 294.3±30.2 | 288.7±29.4 | **357.2±67.2** |
| Medium | 181.9±21.4 | 235.7±33.1 | **343.7±32.7** | 315.5±37.6 |
| Medium Expert | 213.8±57.4 | 274.2±28.5 | 440.9±21.8 | **486.3±41.6** |
| Expert | 277.7±51.7 | 358.2±21.5 | 470.6±21.2 | **487.6±11.8** |

To further show that DOM2 has the ability to generate high-quality actions with policy diversity, we show the number of good policies (i.e., with eposide return higher than 400 in the standard environment) found by the policies in evaluation in Table 8. The results show that under different datasets, DOM2 is able to find more diverse policies than existing algorithms.

Table 8: Comparison of the numbers of good policies found (policies with episode return of the trajectory larger than 400 in the standard environment) in the 3-Agent 6-Landmark environment.

| Dataset | MA-CQL | OMAR | MA-SfBC | DOM2 |
|---|---|---|---|---|
| Random | 0.4±0.5 | 0.8±0.4 | 0.0±0.0 | **12.8±3.4** |
| Random Medium | 0.4±0.8 | 0.6±0.8 | 0.0±0.0 | **9.2±1.6** |
| Medium Replay | 0.2±0.4 | 0.4±0.5 | 4.4±2.8 | **14.6±2.8** |
| Medium | 0.4±0.5 | 0.4±0.8 | 9.6±1.9 | **13.0±2.1** |
| Medium Expert | 0.6±0.8 | 5.2±3.6 | 11.6±1.6 | **19.2±1.2** |
| Expert | 3.8±1.9 | 8.4±2.6 | 17.2±0.8 | **20.0±0.0** |

In spite of the statistical results shown in Table 8, we also visualize the trajectories generated by different algorithms, shown in Figure 10. We rollout the policy trained by different algorithms for 5 times under the same initialized position. The trajectories are colored red, blue, and purple to represent that they are generated by 3 different agents. The green dots are the landmarks. The task for the agents is to reach 3 green landmarks without collision. Different strategies in different background colors mean that 3 agents reach the landmarks and complete the task in various ways. The visualization results show that by multiple attempts, DOM2 finds 5 strategies, however, the numbers of strategies found by MA-CQL, OMAR and MA-SfBC are 2, 2 and 3, respectively. It shows that the policy trained by the DOM2 algorithm possesses high diversity, in other words, DOM2 is capable of generating more diverse policies.

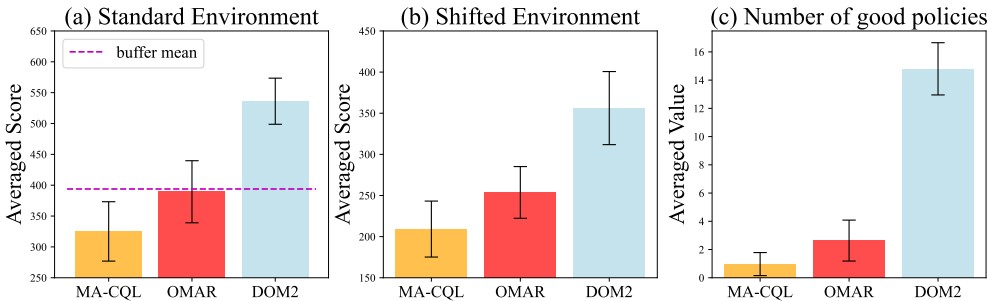

Figure 11: Algorithm performance in 3-Agent 6-Landmark examples among all kinds of datasets. (a): The averaged mean episode returns in the standard environment. (b): The averaged mean episode returns in the shifted environment. (c): Number of good policies (the episode return of the policy more than 400) found in the standard environment.

In Figure 11 (same as Figure 1b), we show the average mean value and the standard deviation value of different datasets in the standard environment as the diagram (a) and in the shifted environment with 10-times evaluation in each episode as the diagram (b). The diagram (c) is the averaged number of good policies (the eposide return is higher than 400 in the standard environment) under different datasets. The performance of DOM2 is shown in the light blue bar. Compared to the MA-CQL as the orange bar and OMAR as the red bar, DOM2 achieves a better average performance in both settings, which means that DOM2 learns policies with much better expressiveness and diversity.

## A.3 Comparison with other Diffusion-based Algorithm

As a different diffusion-based offline MARL algorithm, we compare DOM2 with both MADIFF and DoF along several key dimensions.

- First, DOM2 introduces clearer algorithmic advantages and delivers more reliable performance across diverse tasks and datasets. In contrast to MADIFF's centralized attention-based trajectory diffusion

and DoF's factorized joint-policy diffusion—both of which depend on CTDE and require heavy multi-step generative sampling—DOM2 adopts a fully decentralized few-step policy diffusion with an accelerated solver. This design leads to substantially lower GPU memory usage, faster training and inference, and significantly better scalability in multi-agent scenarios.

- Second, DOM2 integrates diffusion-based policy learning with conservative Q-learning and trajectory-level data reweighting, enabling robust policy improvement beyond the behavior distribution. MADIFF primarily models the dataset distribution without strong corrective mechanisms, making it sensitive to imbalance or low-diversity data, while DoF focuses on trajectory factorization and joint behavior modeling, which can limit robustness and generalization when distributional mismatch occurs.

- Third, DOM2 consistently achieves state-of-the-art performance, strong data efficiency, and superior generalization on both standard and shifted environments. Even in the few settings where its performance is slightly lower, the gap remains small and is mainly attributable to medium-quality datasets that naturally favor trajectory-level diffusion methods like MADIFF and DoF.

Overall, DOM2 offers a more stable, computationally efficient, and broadly generalizable offline multi-agent learning framework than both MADIFF and DoF we next present the experimental results between DOM2, MADIFF and DoF algorithms.

Table 9: Comparison between DOM2, MADIFF and DoF algorithm in the MPE task under different datasets.

| Predator Prey | Random | Medium Replay | Medium | Expert | Average |
|---|---|---|---|---|---|
| MADIFF | 2.0±7.6 | 114.1±17.5 | 142.3±19.7 | 225.2±27.7 | 120.9±18.1 |
| DoF | 24.0±6.1 | 94.0±19.2 | 155.1±18.2 | 223.7±12.0 | 124.2±13.9 |
| DOM2 | **208.7±57.3** | **150.5±23.9** | **155.8±48.1** | **259.1±22.8** | **193.5±38.0** |

| World | Random | Medium Replay | Medium | Expert | Average |
|---|---|---|---|---|---|
| MADIFF | -5.1±2.6 | 42.5±9.2 | **99.0±12.4** | 99.8±3.9 | 59.1±7.0 |
| DoF | 6.2±2.6 | 43.3±9.9 | 67.8±9.1 | **112.6±17.3** | 57.5±9.7 |
| DOM2 | **40.0±14.3** | **65.9±10.6** | 84.5±23.4 | 99.5±17.1 | **72.5±16.4** |

| Cooperative Navigation | Random | Medium Replay | Medium | Expert | Average |
|---|---|---|---|---|---|
| MADIFF | 184.4±11.1 | 268.0±8.9 | **391.5±27.4** | 498.9±18.9 | 335.7±16.6 |
| DoF | 283.0±19.3 | **331.5±12.9** | 375.8±30.3 | 610.7±11.1 | 400.3±18.4 |
| DOM2 | **337.8±26.0** | 324.1±38.6 | 358.9±25.2 | **628.6±17.2** | **412.4±26.8** |

We compare these algorithms across multiple standard tasks and datasets in MPE environments. Across all MPE tasks and dataset types, DOM2 consistently outperforms both MADIFF and DoF, achieving the highest average performance and particularly strong gains on challenging datasets with low data quality. Even in the few settings where DOM2 is slightly behind, the gaps remain small and mainly correspond to Medium datasets, whose balanced quality–diversity structure naturally favors full-trajectory diffusion models.

It is also important to note that MADIFF and DoF rely on CTDE training, granting them access to global information and joint trajectory modeling. In contrast, DOM2 uses a fully decentralized architecture, making the comparison conservative and not strictly fair. Moreover, the public implementation of MADIFF merges multiple datasets, effectively using more training data than DOM2. Despite these advantages for the baselines, DOM2 still provides stronger robustness, scalability, and generalization across all dataset regimes, demonstrating its clear superiority over both MADIFF and DoF.

As shown in Table 10, DOM2 achieves substantially higher computational efficiency than both MADIFF and DoF across all metrics. DOM2 requires only 2.3GB of GPU memory—over 4× lower than MADIFF (10.9GB) and nearly 6× lower than DoF (13.3GB). Its decentralized few-step diffusion policy further reduces

Table 10: Training and inference efficiency comparison via the GPU memory usage, training time (training $10,000$ steps) and evaluating time (summation of evaluating 10 episodes per 100 steps during the process of training 10000 steps) betweeen DOM2 and CTDE-based MADIFF and DoF under the MPE World task using the expert dataset.

| Algorithm | GPU Use | Training time | Evaluation time |
|---|---|---|---|
| MADIFF | 10908MB | 16762.8s | 13123.2s |
| DoF | 13321MB | 15488.8s | 8126.3s |
| **DOM2 (ours)** | **2349**MB | **3307.9**s | **857.2**s |

the training time for 10000 steps to 3307.9 s, representing an $80 - 85\%$ speedup over the CTDE-based MADIFF and DoF baselines. DOM2 also achieves dramatically faster evaluation, completing rollouts in 857.2s, which is roughly $10 - 15\times$ faster than MADIFF and DoF.

These efficiency gains are particularly noteworthy given that MADIFF and DoF rely on centralized training with full global information, whereas DOM2 operates in a fully decentralized setting, making the comparison conservative and inherently favoring the CTDE baselines. Despite this, DOM2 remains the most lightweight and computationally efficient method among all algorithms. Overall, the results clearly demonstrate that DOM2 offers a significantly more scalable, efficient, and practical framework for offline multi-agent RL than both MADIFF and DoF.

## A.4  Further Discussions about Ablation Studies

### A.4.1  Sensitivity Analysis for Trajectory Reweighting Thresholds

To evaluate the impact of different threshold configurations on model performance, we conducted an empirical sensitivity analysis on the Predator Prey task using the Expert dataset. As shown in Table 11, our proposed DOM2 maintains stable and optimal performance across varying thresholds, thereby validating the empirical rationality of our default settings.

Table 11: Performance of DOM2 across varying threshold configurations on the Predator Prey task.

| Threshold Values | 200 | 300 | **400(Ours)** |
|---|---|---|---|
| Performance of DOM2 | $258.4 \pm 53.2$ | $252.5 \pm 28.3$ | $\mathbf{261.5 \pm 36.2}$ |

In practice, identifying a universal set of fixed threshold values is highly non-trivial due to the significant variance in underlying return distributions across different tasks and datasets. As a practical guideline, we select two strategies for candidate threshold selection: (1) setting thresholds based on the quantile distribution of the returns within the specific dataset, or (2) selecting a series of candidate thresholds at uniformly spaced intervals across the effective return range. Exploring adaptive or automated threshold-tuning mechanisms for efficient data reweighting remains a promising direction for future work.

### A.4.2  Ablation of Data Efficiency Without Trajectory Reweighting

To explicitly attribute the source of our observed data efficiency improvements, it is crucial to decouple the contributions of the core diffusion model from the trajectory reweighting mechanism. We clarify that the baseline data efficiency curves presented in the main text for our DOM2 algorithm were evaluated *without* the trajectory reweighting mechanism. This demonstrates that the diffusion model module itself provides a strong inherent contribution to the sample efficiency. To further illustrate the additional performance gains provided by the reweighting mechanism, we conducted supplementary evaluations on the Predator Prey task under varying sample sizes, as detailed in Table 12.

The results show that the base DOM2 without reweighting already establishes a highly data-efficient baseline that significantly outperforms prior methods such as MA-CQL, OMAR, and MA-SfBC. When the data

Table 12: Data efficiency evaluation on the Predator Prey task across varying numbers of samples.

| Number of Samples | $1 \times 10^5$ | $5 \times 10^5$ | $1 \times 10^6$ |
|---|---|---|---|
| MA-CQL | $0.589 \pm 0.164$ | $0.585 \pm 0.159$ | $0.510 \pm 0.204$ |
| OMAR | $0.761 \pm 0.162$ | $0.739 \pm 0.197$ | $0.736 \pm 0.184$ |
| MA-SfBC | $0.872 \pm 0.187$ | $0.865 \pm 0.145$ | $0.893 \pm 0.160$ |
| DOM2 w.o. Reweighting | $0.953 \pm 0.154$ | $0.997 \pm 0.150$ | $1.066 \pm 0.170$ |
| **DOM2** | $\mathbf{0.954 \pm 0.197}$ | $\mathbf{1.021 \pm 0.238}$ | $\mathbf{1.127 \pm 0.177}$ |

reweighting mechanism is enabled, both the overall performance and data efficiency are further amplified. By comparing these settings, we clearly delineate our sources of improvement: the core diffusion model establishes a robust baseline, while the data augmentation technique serves as an effective performance amplifier.

