# OpenReview forum: "Improving Generalization and Data Efficiency with Diffusion in Offline Multi-agent RL"
_TMLR — Accepted by TMLR_

### Review · Reviewer_WkRB · 2025-12-03

**Summary Of Contributions:**

**Paper summary**

The paper proposes a novel diffusion multi-agent offline reinforcement learning model (DOM2). The contributions in their methodology are threefold:

- diffusion: generating actions by denoising a Gaussian noise.
- policy improvement: optimizing both the Q-loss and the diffusion loss.
- data reweighting: scaling up the training data by replicating trajectories with high return values.

Through experiments on two environments (MPE and MAMuJoCo), 6 datasets (random, random-medium, medium-replay, medium, medium-expert, expert), and 4 offline multi-agent RL baselines (MA-CQL, OMAR, MA-SfBC, MA-DIFF), they demonstrate that:

- their method outperform all baselines on all environments and almost all data setups.
- their methods generalizes better to shifted environments (designed by changing some environment parameters such as the speed of agents in the MPE environment).
- their method is more data efficient. I.e. they were able to achieve the same baseline performance using only 5% of the data.

**Strengths**

- They conduct extensive experiments comparing their method to other offline multi-agent RL baselines.
- They highlight the key differences in methodology between their method and other multi-agent diffusion-based RL methods.

**Weaknesses**

- In the experiment setup section (5.1) you describe that you compare to 4 different baselines, including MA-DIFF, however, it is not included in the main results in tables 1, 2, and 3 and no clarification or justification is mentioned.
- As the data quality increases, the gap between the diffusion based algorithms decreases, as can be seen in Figure 5(d). No discussion or comment on this behaviour has been made. Are the benefits of DOM2 more pronounced in random data scenarios?

*Disclaimer: I do not have a strong experience on the specific area so my comments are mostly regarding the experiment design and results presented.*

**Audience:**

Yes

**Audience Explanation:**

The use of diffusion in offline multi-agent RL systems is a useful idea to the audience interested in offline RL systems.

**Claims And Evidence:**

Yes

**Claims Explanation:**

Main claims are supported by extensive experiments. They also include ablations to justify the importance of each component in their algorithm.

**Requested Changes:**

- **Critical**
    - figure 1c referenced in the caption of Figure 1 is missing.
    - ensure consistency of the text and the results presented.
    - elaborate on the discussion of the correlation between performance and data quality.

---

> ### Author Response · Authors · 2025-12-05
> **Response to Reviewer WkRB**
>
> We sincerely thank the reviewer for the constructive comments and positive assessment of our work. Below we address the raised concerns.
>
> **1. The comparison of MA-DIFF as the baseline in the main result tables**
>
> Thank you for pointing this out. Although MA-DIFF is listed among the baselines in Section 5.1, differences in task settings and dataset construction make its results not directly comparable to DOM2 in the main experiments. Therefore, to ensure fairness, we present MA-DIFF results in the Section 5.4 and Appendix A.3, where the setting differences are clearly explained. Please refer to Section 5.4 (Table 4) and Appendix A.3 (Table 9) for details. Results show that DOM2 has superior performance compared with other baseline algorithms. We will update the main text to clarify this and maintain consistency.
>
> **2. On the decreasing performance gap with sub-optimal baseline algorithm**
>
> We appreciate this insightful observation. As shown in Figure 5(d), the performance gap between DOM2 and other diffusion-based baselines indeed narrows with more high-quality data. This aligns with the design of DOM2:
>
> DOM2 performs explicit policy improvement, whose effectiveness depends on accurate Q-value estimation. More data leads to more reliable Q-values and thus stronger policy improvement.
>
> Behavior-cloning-based diffusion baselines rely mainly on data distribution, which changes less dramatically with increased dataset size.
>
> Therefore, DOM2 shows the greatest advantage in low-quality or low-data regimes, while the gap naturally decreases as data quality improves.
>
> More generally, in offline RL, algorithm performance is inherently tied to dataset quality due to the absence of environment interaction. We will add this discussion to the revised paper.
>
> **3. Missing Figure 1(c) and consistency**
>
> We thank the reviewer for identifying this issue. Figure 1(c) was omitted during compilation, and we will restore it and ensure all references and results are consistent.
>
> We thank the reviewer again for the valuable feedback, which has helped us improve the clarity and completeness of the paper.

---

> ### Comment · Reviewer_WkRB · 2026-04-01
> **No response from authors**
>
> Dear authors,
> Can you please respond to the questions and weakness points raised in my review?

---

> ### Author Response · Authors · 2026-04-01
> **Response**
>
> Dear Reviewer,
>
> Thank you for your comment and for your patience. We would like to clarify that we submitted our response promptly after receiving the review (05 Dec 2025, 21:03). However, due to a system-related issue, the submission interface at that time did not provide an option to make the response visible to reviewers.
>
> Furthermore, after all reviewers had submitted their comments, the system did not update accordingly, which unfortunately resulted in our response not being visible.
>
> We have now updated the response to ensure it is accessible. We sincerely apologize for any confusion this may have caused and kindly invite you to check again. We are, of course, happy to further address any questions or concerns you may have.
>
> Best regards,
> The Authors

---

> > ### Comment · Reviewer_WkRB · 2026-04-07
> > **Response**
> >
> > Thank you for the clarifications. My concerns have been addressed.

---

### Review · Reviewer_57L6 · 2025-12-05

**Summary Of Contributions:**

This paper focuses on the tasks of offline multi-agent reinforcement learning. While current methods, like SAC, tend to be conservative when handle extrapolate scenarios, the authors introduce a diffusion model to replace the role of the behavior policy (actor), which provides more diversity for action planning. They also introduce a Q-loss as a regularization term in the training process and a data-reweighting mechanism during dataset preparation. As a result, their method outperforms current offline MARL methods in average scores, convergence rate, and GPU cost.

### Strengths
* The writing is fluent and the narrative is complete. The overall paper is easy to understand.
* The experiments are solid and the results are promising, especially in shifted-environment cases.
* Not only does DOM2 outperform baselines after training, but it also provides higher convergence rates with less GPU memory (compared with other diffusion methods), which is important for RL since the scarcity of data samples is always a problem when training an RL model in real-world scenarios.

### Weaknesses
1. The paper does not show why this method is particularly good for multi-agent RL. It seems also work for single-agent RL. For example, in the loss design, each agent only has its own loss; there is no consideration of the global environment or impact from other agents.
2. The data reweighting part is too concise and does not describe how the thresholds are derived. Besides the fact that the experimental results are better, the motivation is not well stated. According to the statement: *“Doing so allows us to create more data efficiently and improve the performance of the policy by increasing the probability of sampling trajectories with better performance in the dataset.”*, it gives the impression that only good trajectories are needed for training. Then why not just keep trajectories with high returns and delete all trajectories with low returns? Would the performance be better in that case?
3. The comparison with other diffusion methods should be more detailed and complete. Only one diffusion-related RL method (DoF) that proposed later than 2023 is mentioned. More recent papers are not covered, e.g., Diffusion Actor-Critic [1].

[1] Fang, Linjiajie, et al. "Diffusion actor-critic: Formulating constrained policy iteration as diffusion noise regression for offline reinforcement learning." arXiv preprint arXiv:2405.20555 (2024).

**Additional Comments:**

Additional questions that I am curious about:

1. Interested in the results for the single-agent setting compared with other single-agent SOTA methods.
2. Why do we need to compare different levels of dataset quality as shown in the tables?
3. It would be better to add an ablation study to show the contribution of each component to the convergence rate, not just the scores.
4. What is the behavior policy in your paper? The ground truth provided by the dataset/environment, or another trained network?

**Audience:**

Yes

**Audience Explanation:**

**Timeliness:** The intersection of diffusion models and reinforcement learning is currently one of the most active research areas in machine learning. This paper provides a compelling and complete system that build the policy model with different neural network framework, insightful for those who are familiar with RL but not with Diffusion.

**Practicality:** The “ultra data efficiency” finding (20× improvement) is of significant interest to practitioners in robotics and autonomous systems where data collection is expensive.

**Problem Solving:** The paper addresses a well-known limitation in offline RL—the “conservatism” trap where agents fail to cooperate because they are penalized for deviation—and offers a novel solution via generative modeling.

**Broader Impact Concerns:**

No ethical concerns.

**Claims And Evidence:**

Yes

**Claims Explanation:**

The authors make four primary claims:
(1) DOM2 enhances policy expressiveness/diversity,
(2) it improves robustness against environment shifts,
(3) it improves data efficiency, and
(4) it outperforms current methods.

The evidence provided to support these claims is empirical and generally convincing.

**Agree:**
* **Performance (Claim 4):** The paper provides extensive comparison tables (Table 1, Table 3, Table 4) showing DOM2 consistently achieving higher average scores and lower variance compared to strong baselines like MA-CQL and OMAR across multiple datasets (MPE and MAMuJoCo).
* **Generalization (Claim 2):** The “Shifted Environment” experiments are strong evidence. By altering agent speeds (MPE) or physics parameters (MuJoCo) during evaluation, the diffusion-based policy retains performance while conservative baselines collapse (Table 2, Table 3). This strongly supports the hypothesis that diffusion captures a more diverse set of effective behaviors.
* **Data Efficiency (Claim 3):** Figure 5 and Table 10 explicitly plot performance against dataset size, showing DOM2 achieving near-optimal returns with only 5% of the data ($5 \times 10^4$ samples).
* **Ablation (Mechanism Support):** Figure 6 isolates the contributions of the diffusion loss, Q-loss, and data reweighting, confirming that each component is necessary for the reported gains.

**Partially Agree:**
* **Diversity (Claim 1):** The sanity case in Figure 1 and Figure 10 shows that DOM2 provides diverse solutions across rollout rounds, but this diversity is not showcased clearly in the main experiments.
* **Expressiveness (Claim 1):** The terminology is unclear (at least for me), and there is insufficient discussion of this aspect.

**Requested Changes:**

1. Comparison with other diffusion methods should be more detailed and complete. More recent papers are not covered, as I mentioned in summary.
2. Figure 5(d) does not show the curve for MA-SfBC.
3. Typos:
   * “unseened actions” → “unseen actions”.
   * “Cooperivate Navigation” (Fig. 5(c), 6(c)) → “Cooperative Navigation”.
   * “Standard Brownion motion” → “Brownian motion”.

---

> ### Author Response · Authors · 2026-02-02
> **Response to Reviewer 57L6**
>
> Thank you for the insightful comments. We address the concerns point-by-point below.
>
> (1) Why this method is particularly good for multi-agent RL
>
> Our method is designed specifically for the challenges of offline multi-agent RL, where non-stationarity, partial observability, and heterogeneous behavior policies make policy learning substantially harder than in single-agent settings. To address these issues, we adopt a fully decentralized training framework, allowing each agent to learn a policy and critic that remain stable even when other agents’ behaviors vary to improve policy generalization and data efficiency.
>
> Our algorithm is explicitly tailored for multi-agent settings, as evidenced by our use of aggregated global rewards for Q-function training and agent-specific data augmentation that addresses the divergence between local and global reward perspectives as a design specificity. Although the conservative objective originates from single-agent RL, it serves a unique purpose in MARL: each agent faces significant distribution shift because its transition dynamics implicitly depend on other agents’ actions. The conservative critic mitigates this non-stationarity by preventing overestimation under multi-agent data distributions, enabling reliable policy updates in the offline regime, especially reliable in multi-agent settings.
>
> By combining decentralized learning, diffusion-based policy modeling, and conservative value estimation, our approach is particularly effective for multi-agent systems, which is reflected in its strong empirical performance across diverse MARL benchmarks.  We will further clarify in the revised manuscript.
>
> (2) Motivation and design of the data-reweighting thresholds
>
> Our goal is not to keep only high-return trajectories. Removing all low-return data may affect learning in two ways: (i) it removes negative examples that are essential for stabilizing Q-learning, and (ii) it creates severe distribution shift because the remaining data no longer covers the state–action space enough. In practice, this often leads to overestimation and poor generalization.
>
> Therefore, instead of hard filtering, we adopt soft reweighting, which increases the sampling probability of high-quality trajectories while still preserving coverage. Given that the minimum return $R_{min}$ and maximum value $R_{max}$ in the dataset.
>
> The thresholds are derived directly from the return distribution:
>
> 1.Quantile-based thresholds to adapt to skewed datasets (e.g., median, 3/4, 7/8 quantiles and , 15/16 quantiles as$[R_{0.5},R_{0.75},R_{0.875},R_{0.9375}]$, here $R_{percent}$ means that if we randomly sample a trajectory in the dataset and the return is $r,P(r<=R_{percent}=percent))$. We use this threshold in MPE World medium-replay and medium dataset, Cooperative Naviagtion medium-expert dataset and HalfCheetah medium-expert dataset.
>
> Range-based thresholds by evenly partitioning $[R_{min},R_{max}]$ to maintain coverage across different return levels (e.g., $[R_{min},R_{min}+C，R_{min}+2C,...,R_{max}-C,R_{max}],R_{min}=0,R_{max}=200,C=10$). In spite of the abovementioned and random datasets, other datasets all use this method as the data reweighting.
>
> This provides a controllable way to bias training toward better trajectories without losing necessary low-return data.
>
> We will include the “high-return-only” filtering as an explicit baseline in the revision to demonstrate the difference. The results show that removing the low-quality datasets leads to a performance degradation due to distribution shift.
>
> | Task                   | Use Full Data | Use the First 1% Good Data |
> |------------------------|---------------|----------------------------|
> | Predator Prey          | 155.8 ± 48.1  | 138.9 ± 53.4               |
> | World                  | 84.5 ± 23.4   | 64.5 ± 22.1                |
> | Cooperative Navigation | 358.9 ± 25.2  | 304.5 ± 20.4               |
>
> (3) Comparison to other diffusion-based RL methods (e.g., DAC)
>
> We appreciate the suggestion and will expand the discussion of diffusion-related RL methods. The Diffusion Actor-Critic (DAC) paper provides an interesting advancement, and it is fundamentally designed for single-agent offline RL with constrained policy iteration, and its objective differs from DOM2. DAC introduces a score-matching term involving score function differences and Q-function gradients, which follows from its constrained optimization formulation.
>
> Our work focuses on multi-agent offline RL and our objective is to directly maximize the expected episodic return. Accordingly, our loss consists of a policy improvement term combined with a behavior cloning loss.
>
> We will add a detailed discussion on this distinction in the revised version.

---

> ### Author Response · Authors · 2026-02-02
> **Response to Reviewer 57L6 (Continued)**
>
> For diversity and expressiveness (Claim 1), our intention is to clarify the following:
>
> By expressiveness, we refer to the representational capacity of the policy network. A standard MLP-based policy typically outputs the parameters of a Gaussian distribution, which limits the range of behaviors it can model. In contrast, a diffusion-based policy can represent much richer and more flexible action distributions. This higher expressiveness enables the policy to capture multimodal behaviors and potentially discover more feasible solutions in complex environments.
> The diversity observed in the sanity-check examples (Figure 1 and 10) is a direct consequence of this stronger expressiveness. While the main benchmark tasks naturally constrain the range of optimal behaviors (making diversity less visually salient), the improved expressiveness still contributes to better performance. We will clarify these points and more explicitly connect expressiveness and diversity in the revision.
>
> Response to the Requested Changes:
>
> Thank you for pointing this out. We will include a more complete discussion comparing our method with other recent diffusion-based approaches and expand the related-work section accordingly. We also appreciate your careful reading of the paper and the typos you mentioned will be corrected in the revision.
>
> Response to Additional Comments:
>
> Our method is primarily designed to address policy generalization and data-efficiency challenges in offline multi-agent RL, where the problem is substantially more difficult and decentralized training provides better scalability. In the single-agent setting, our architecture is similar to Diff-QL and SfBC in using diffusion models as policies, but differs in using CQL to train the Q-function. This improves robustness when data is scarce, which explains our strong performance on the benchmark tasks—though such comparisons are not entirely fair, as our method is motivated by the multi-agent setting.
> Comparing across different dataset quality levels is standard in offline RL because policy performance is tightly coupled with the quality of the data; evaluating on multiple datasets of varying quality is therefore necessary to provide a complete and fair assessment.
> The contribution of each component to performance has been analyzed in detail in Section 5.4 (Figure 6), where we provide ablations for the main components.
> The behavior policy in our experiments refers to the policy used to collect the dataset. We follow the OMAR[1] benchmark and use the same datasets to ensure fairness and consistency in comparison.
>
> [1]Plan Better Amid Conservatism: Offline Multi-Agent Reinforcement Learning with Actor Rectification. ICML2022

---

### Review · Reviewer_PfVB · 2026-02-23

**Summary Of Contributions:**

The paper proposes DOM2, a diffusion-based policy framework for offline multi-agent reinforcement learning under fully decentralized training and execution. The method combines conditional diffusion policies trained via score matching with Q-guided improvement using a CQL critic, and introduces a trajectory-level data reweighting scheme. Experiments on MPE and MAMuJoCo show strong performance across dataset qualities, improved robustness under environment shifts, and reported gains in data efficiency.

Strengths:  1) Strong experiment setting and results, 2) evaluation under shifted environments, 3) computational benefits of decentralized diffusion design.

Weaknesses: 1) Lack of analysis for trajectory reweighing thresholds, 2) not clear which component is responsible for the reported data efficiency improvement.

**Audience:**

Yes

**Audience Explanation:**

Yes. The integration of diffusion models into offline multi-agent RL addresses an active research direction, and their strong empirical performance and robustness results are of interest to researchers working on offline RL and multi-agent systems.

**Broader Impact Concerns:**

No ethical concerns.

**Claims And Evidence:**

Yes

**Claims Explanation:**

The paper supports their main claims about performance and robustness through extensive experiments on MPE and MAMuJoCo tasks across multiple dataset qualities from random to expert. It evaluates standard and shifted environments, comparing against offline MARL baselines, and includes ablation studies removing diffusion loss, Q-guidance, and data reweighting. The reported performance improvements and computational efficiency claims are generally well supported by their results. However, the claim of significant data efficiency improvement (20×) is not fully isolated from the trajectory reweighting strategy, and no sensitivity analysis is provided for the reweighting thresholds. Additional ablation would strengthen the experiments.

**Requested Changes:**

1) Sensitivity analysis for trajectory reweighting thresholds: Providing empirical sensitivity analysis or guidance for selecting the return thresholds used in the trajectory reweighting would strengthen the paper.
2) Ablation of data efficiency without trajectory reweighting: The authors should evaluate whether the reported data efficiency gains hold when trajectory reweighting is disabled, to clearly attribute the source of these improvements.

---

> ### Author Response · Authors · 2026-03-02
> **Response to Reviewer PfVB**
>
> We sincerely thank the reviewer for the thoughtful and constructive feedback. We appreciate the insightful comments and address each of the questions in detail below.
>
> Question1: Sensitivity analysis for trajectory reweighting thresholds: Providing empirical sensitivity analysis or guidance for selecting the return thresholds used in the trajectory reweighting would strengthen the paper.
>
> Answer1: We sincerely thank the reviewer for this constructive suggestion. We strongly agree that providing further insights and guidance on selecting the trajectory reweighting thresholds strengthens the paper.
> To address this, we have made the following updates to our manuscript:
> 1. Empirical Sensitivity Analysis: In our ablation study, we have highlighted the overall impact of the data reweighting module. Furthermore, we have added a new case study on the Predator Prey task using the Expert dataset and the results are show in the table below. We evaluated the model's performance across varying threshold configurations. The results demonstrate that our chosen thresholds yield stable and optimal performance, validating the empirical rationality of our settings.
>
> | Threshold Values  |200|300|400|
> | ----------- | ----------- |-----------|-----------|
> |Performance of DOM2|258.4$\pm$ 53.2|252.5$\pm$ 28.3|261.5$\pm$ 36.2|
>
> 2. Guidance for Threshold Selection: We have added a discussion regarding how to select these thresholds in practice. Since the underlying return distributions vary significantly across different tasks and datasets, identifying a universal set of fixed threshold values is highly non-trivial. The dimensionality of the threshold search space also makes exhaustive ablation computationally prohibitive. As a practical guidance, we select two strategies for selecting candidate thresholds: (1) setting them based on the quantile distribution of the returns in the specific dataset, or (2) selecting a series of candidate thresholds at uniformly spaced intervals across the effective return range.
>
> We have included this discussion in the revised manuscript and explicitly highlighted that exploring adaptive or automated threshold-tuning mechanisms for efficient data reweighting/augmentation is a promising direction for future work.
>
>
> Question2: Ablation of data efficiency without trajectory reweighting: The authors should evaluate whether the reported data efficiency gains hold when trajectory reweighting is disabled, to clearly attribute the source of these improvements.
>
> Answer2: We sincerely thank the reviewer for this insightful comment. We completely agree that clearly attributing the source of the data efficiency improvements is crucial for validating our method.
> To address this, we would like to clarify the existing results and present our newly added evaluations:
>
> 1. Baseline Data Efficiency (Without Reweighting): We would like to clarify that the data efficiency curves for our DOM2 algorithm, as originally presented in the "Data Efficiency" section of our manuscript, were actually evaluated without the trajectory reweighting (data augmentation) mechanism. This confirms that the reported data efficiency gains indeed hold when reweighting is disabled, demonstrating that the diffusion model module itself provides a strong inherent contribution to the sample efficiency.
>
> 2. Enhanced Data Efficiency (With Reweighting): To provide a comprehensive ablation and explicitly illustrate the additional contribution of the trajectory reweighting mechanism, we have conducted new experiments on the Predator Prey task via different datasets. We add the table below of the normalizing scores to show the superior performance of our algorithm.
>
> | Number of Samples  | $1\times10^5$|$5\times10^5$| $1\times10^6$|
> | ----------- | ----------- |-----------|-----------|
> |    MA-CQL   |    0.589$\pm$0.164   |0.585$\pm$0.159|0.510$\pm$0.204|
> | OMAR   | 0.761$\pm$0.162 |0.739$\pm$0.197|0.736$\pm$0.184|
> | MA-SfBC   | 0.872$\pm$0.187  |0.865$\pm$0.145|0.893$\pm$0.160|
> | DOM2 w.o. Reweighting   |    0.953$\pm$0.154     |0.997$\pm$0.150|1.066$\pm$0.170|
> | **DOM2(Ours)**   | 0.954$\pm$0.197 | 1.021$\pm$0.238 | 1.127$\pm$0.177|
>
>
> By comparing the newly added curves (with reweighting) against the original ones (without reweighting), we can clearly delineate the sources of our improvements: the core diffusion model establishes a highly data-efficient baseline, while the data reweighting mechanism further amplifies both the overall performance and the data efficiency. We have updated the text in the revised manuscript to make this distinction explicitly clear.

---

### Comment · Reviewer_57L6 · 2026-02-02
**Checklist for revision**

Hi Authors,


I noticed that you submitted a revision. Could you please either:
1. highlight the part you edited in the revised version of paper

or

2. write a checklist in Openreview so that we can quick check what are different.

or

3. reply the points you addressed in latest version under per reviewers' space with detailed.

Thank you!

---

### Decision · Action_Editor_aait · 2026-05-07

**Recommendation:** Accept with minor revision

**Audience:**

Yes

**Audience Explanation:**

Researchers on reinforcement learning or diffusion models will be interested.

**Claims And Evidence:**

Yes

**Claims Explanation:**

The paper's core claims: superior performance, generalization under environment shifts, and data efficiency, are well supported by extensive experiments across MPE and MAMuJoCo benchmarks with multiple dataset quality levels. Ablation studies (Figure 6) isolate the contribution of each component (diffusion loss, Q-guidance, data reweighting). During the review process, the authors further strengthened the evidence by providing (1) a sensitivity analysis for reweighting thresholds showing stable performance across configurations, (2) an ablation confirming data efficiency gains hold even without reweighting, clearly attributing the improvement to the diffusion model itself, and (3) a hard-filtering baseline demonstrating that soft reweighting outperforms naive trajectory removal. All three reviewers agreed the claims are supported, and the remaining concern regarding expressiveness/diversity (Reviewer 57L6) is minor and pertains more to terminology clarity than empirical validity.